# A multi-host mechanistic model of African swine fever emergence and control in Romania

Brandon Hayes [1,2] ✉, Timothée Vergne [1,4], Nicolas Rose[2], Cristian Mortasivu[3] & Mathieu Andraud [2,4]

The on-going African swine fever pandemic has been devastating to affected nations, with continued spread observed despite aggressive interventions. Suspected interspecific transmission among wild and domestic hosts further complicates control efforts, yet its role in epidemic propagation remains poorly understood. Here, we develop and calibrate a multi-host mechanistic transmission model to the first wave of the epidemic in Romania (June–December 2018), quantifying these dynamics and evaluating counterfactual management scenarios. We estimated that 60% (95% credible interval: 27–83%) of outbreak farms were linked to other outbreak farms, 27% (5.3–67%) to infected wild boar populations, and 13% (1.9–27%) to external sources. For wild boar, 39% (3.8–93%) of infected populations were estimated to have originated from outbreak farms, and 61% (7.3–96%) from other infected wild boar populations, with favorable habitat exhibiting higher susceptibility and infectivity than unfavorable habitat. Among alternative control strategies, reactive and preventive culling of domestic pig herds yielded the greatest decrease in median final epidemic size among domestic pigs. These findings provide quantitative evidence that interspecific transmission was a critical epidemic driver, and a necessary target for achieving holistic control. Our model offers a flexible, rapidly-deployable framework for informing surveillance and response policy in at-risk regions.

African swine fever (ASF), a high-consequence disease of domestic pigs, continues to threaten the global swine industry[1]. Virus introduction and transmission pathways vary among affected areas[2]: In the Baltic nations[3] and the Republic of Korea[4], cases have been seen to occur predominantly or exclusively among wild boar, while in Romania[5], Ukraine[6], China, and southeast Asian nations[7] domestic pig cases predominate. In countries with cases in both compartments[2,8], interspecific transmission is suspected, but surveillance biases hinder accurate evaluation of this transmission route[7]. With no available treatment and an effective vaccine still out-of-reach, interrupting disease transmission remains dependent on control strategies that target these pathways[9,10]. Understanding the dynamics unique to each epidemic provides the best chance for achieving epidemic control[11].

Control measures against ASF are encoded in European Union (EU) legislation, aligned with World Organization for Animal Health (WOAH) international standards and are to be applied anywhere ASF is suspected or confirmed[12]. While the 2023 adoption of special legislation has harmonized management approaches, control strategies previously—at the time of the epidemic under study—varied across Member States[13]. For domestic pigs, such measures consisted of

[1]Univ Toulouse, ENVT, INRAE, IHAP, Toulouse, France. [2]Ploufragan-Plouzané-Niort Laboratory, the French Agency for Food, Environmental and Occupational Health & Safety (ANSES), Ploufragan, France. [3]Romanian National Veterinary Sanitary and Food Safety Authority (ANSVSA), Bucharest, Romania. [4]These authors jointly supervised this work: Timothée Vergne, Mathieu Andraud. ✉e-mail: brandon.hayes@envt.fr

stamping out affected farms, performing animal and animal product contact tracing, and establishing protection and surveillance zones around affected premises to enable targeted disinfection, movement restrictions, and active surveillance measures[14]. For wild boar, the recommended strategic approach to ASF focused on establishing core infected and surrounding surveillance zones in which active carcass search and removal, installation of fences, and intensive wild boar depopulation were considered[15].

Mechanistic models—which allow the quantification of transmission parameters, the prediction of epidemic trajectories, and the evaluation of control strategies—are a proven way of providing quantitative information to guide potential control measures[16–19]. However, ASF models addressing transmission between domestic pigs and wild boar remain limited in number and are not parameterized using empirical outbreak data, despite evidence suggesting that domestic-wildlife interactions contribute to epidemic propagation[20]. An improved understanding of the transmission dynamics at this interface, adapted to the specificities of domestic pig rearing according to local socio-economic determinants, is necessary for designing tailored control measures.

Romania has been experiencing an ASF epidemic of unprecedented scale since 2018. Genotype II ASF virus (ASFV) strains circulating among wild boar and domestic pigs[5] have been linked to isolates from other EU Member States (Lithuania and Poland) and the Caucasus region[21]. The ubiquity of backyard pig farming in villages—where pigs are kept by nearly every household in low-biosecurity, small-scale holdings—has created favorable conditions for sustained transmission at the domestic-wildlife interface[22]. Gaining insight into the dynamics during the early stages of the epidemic may support more tailored approaches to control, particularly in regions not yet affected by ASF.

To investigate these dynamics, we focused on the initial wave of the current epidemic, from June to December 2018, that exhibited clearly-defined onset, rapid growth, and recovery phases[22]. Occurring across six counties in southeastern Romania (Braila, Calarasi, Constanta, Galati, Ialomita, and Tulcea), this transmission period was distinct from the subsequent wave that began in early 2019, and was aligned with the backyard pig-rearing production dynamics that begin in the late spring and end during the holiday slaughter in mid-December[23]. Control measures implemented during this period were in accordance with EU directives, but also adapted to local production dynamics and socioeconomic acceptability[23,24]. Following the culling of an infected farm, a 10km surveillance zone would be established around the premises and maintained for 4 weeks, provided no additional cases were detected. However, while all herds in a commercial farm were depopulated, backyard farms were considered individually: neighboring backyard herds were not automatically culled despite contact and transmission likelihoods. Further surveillance and response heterogeneity among wild boar was reported between counties.

Using a multi-host spatiotemporal model of ASF transmission calibrated to the observed emergence dynamics, we aimed to estimate the relative contribution of domestic pigs and wild boar to overall epidemic propagation, identify and quantify the parameters needed to explain the observed epidemic trajectory, and evaluate potential outcomes from alternative control strategies, had they been implemented during this period of epidemic onset.

## Results
### Model design
Our model sought to capture the spread of ASF in a multi-host system—consisting of backyard domestic pig farms and patches of wild boar populations in Romania—by considering the transmission dynamics within and between host groups. The model was calibrated to weekly case detection data for outbreaks among pig farms and detections of wild boar carcasses. A total of 256 model formulations were evaluated, representing all combinations of structural formulation variations, to identify the model frameworks, transmission formulations and contact structures that best captured the epidemic. The best-fitting model was defined as the first among the top-ranked candidate models (based on the lowest total distance to the observed summary statistics) where the observed weekly incidence of the majority of counties fell entirely within the 95% credible intervals of the model's posterior predictions. Parameter values of this model were then used to evaluate the outcomes of alternative control strategies—improved passive surveillance, reactive village-wide culling, preventive culling, and environmental sanitation of infected wild boar carcasses—had they been implemented during this period of epidemic onset.

### Model structure
Wild boar case detection was best explained using forest coverage as a habitat proxy, having outperformed mean wild boar density, with an area under the curve (AUC) of 0.86 compared to 0.68 (Supplementary Fig. 1). A threshold of 10.5% forest coverage (sensitivity 74.5%, specificity 83.7%) was identified through receiver operating characteristic (ROC) analysis that compared forest coverage and wild boar density as wild boar case predictors. Forest coverage was therefore used to define suitable wild boar habitat in model construction.

The influence of each structural parameter was quantified through permutation feature importance. The transmission mode for transmission between domestic pig farms was seen to be most influential (Supplementary Fig. 2), with a frequency-dependent transmission formulation consistently associated with better-fitting models than a density-dependent formulation (Supplementary Fig. 3). The contact structure between wild boar habitat patches and farms was the next most-influential structural choice, with models that assumed direct (zero-order) contact outperforming those that included first-order neighbor contacts. The wild boar-to-farm transmission mode had minimal impact.

### Model fitting and inference
The best fitting model contained 11 parameters, having utilized farm-to-farm transmission rates stratified by county for high- medium- and low-incidence counties, uniform wild boar patch-to-patch transmission rates across counties, no explicit external force of infection for wild boar, zero-order contact from patches to farms and farms to patches, frequency-dependent transmission from farms to farms and farms to patches, and density-dependent transmission from patches to farms. Visual inspection of the best-fitting model's 95% prediction intervals confirmed that the weekly incidence and spatial distribution across counties were captured for each host (Fig. 1).

Parameter posterior estimates, obtained through a sequential Monte Carlo approximate Bayesian computation algorithm, are presented in Table 1 with posterior-prior comparisons available in the supplemental material (Supplementary Fig. 4). A total of 2600 simulations were required to achieve a final posterior sample size of 100 particles. Certain parameters, such as the relative infectivity of farms, were seen to be only weakly informed by the data, while others, like the relative susceptibility of wild boar patches or the farm-to-farm transmission rates, were more clearly identifiable. Model predictions against the data, at the level of the summary statistics used for fitting, are presented in the supplementary material (Supplementary Fig. 5).

### Infection source estimation
Using the best-fitting model, the proportional contributions of different transmission pathways to overall epidemic dynamics were estimated (Fig. 2) and summarized by their posterior mean and 95% credible interval. For domestic pigs, a posterior mean of 60% (95% credible interval: 27–83%) of infections originated from other farms, 27% (5.3–67%) came from infected wild boar habitat patches, and 13%

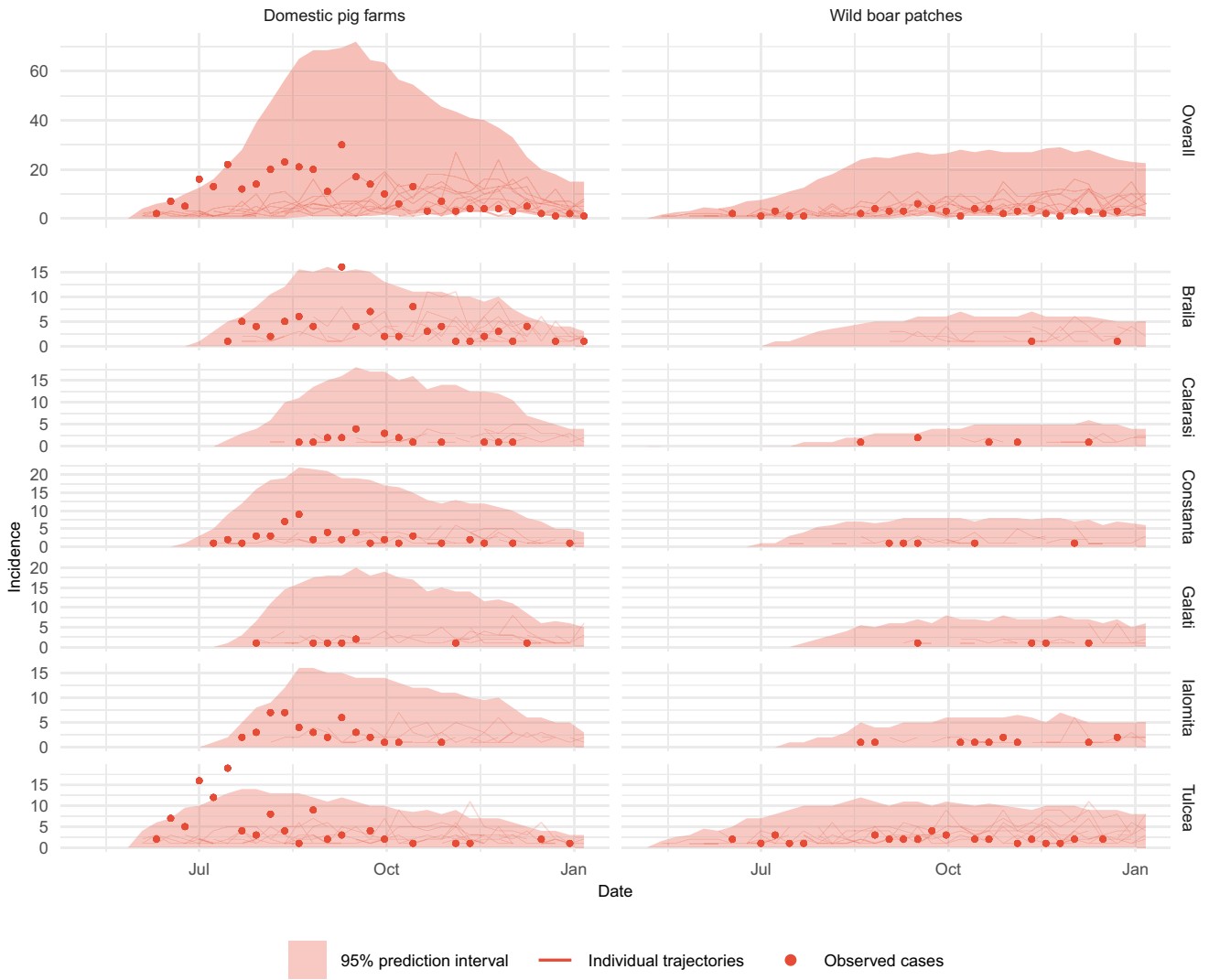

**Fig. 1 | Observed and predicted weekly incidence in domestic pig farms and wild boar patches across six counties in southeastern Romania from June to December 2018.** Shaded ribbons represent 95% prediction intervals of the best-fitting model (determined across 500 simulations of 100 conserved particles), points indicate observed weekly case counts, and thin lines show trajectories from the ten top-ranked stochastic replicates selected using residual mean squared error. The top row shows the aggregate model fit across all counties by epidemiological unit, while the lower panels show incidence further stratified by county.

(1.9–27%) from external sources. Conversely, for wild boar patches, 39% (3.8–93%) of cases came from farms and 61% (7.3–96%) from other patches. A graphical summary of these results is available in the supplementary material (Supplementary Fig. 6). When faceted by county, similar patterns were seen (Supplementary Fig. 7). Wild boar habitat with at least 10.5% forest coverage was estimated to be 3.4 (1.2–125) times more infectious and 34 (1.5–500) times more susceptible than those with lower coverage.

### Alternative management scenarios
Relative to the baseline scenario, preventive and reactive culling of domestic pig farms led to the largest median reductions in final epidemic size among farms, by 97 (scenario-baseline 95% credible interval: −715 to 558) and 73 (−706 to 572) cases, respectively. However, Bayesian comparison of final epidemic sizes revealed weak evidence for these strategies, with probabilities of achieving a smaller epidemic than that of the baseline estimated at only 61% and 59% for preventive and reactive culling (Table 2). More modest reductions in epidemic size were observed for scenarios of environmental sanitation and improved passive surveillance, with similar uncertainty profiles for their effectiveness.

### Performance-weighted ensemble modeling
To evaluate how alternative model structures compared to the selected best-fitting model, the models that met the inclusion criteria were ranked by their final Euclidean distance, and a performance-weighted ensemble from the models in the lowest-distance cluster was created (Supplementary Fig. 8). While three clusters emerged numerically, the models in the two lowest-distance clusters produced near-identical trajectories (Supplementary Fig. 9) having shared key influential features (frequency-dependent farm-to-farm transmission mode, zero-order patch-to-farm and farm-to-patch contact structure, and no external force of infection on wild boar patches). Conversely, models in the third cluster demonstrated increasingly divergent behavior, particularly among the wild boar component (Supplementary Fig. 9). The first two numerical clusters were thereby considered as a single cluster of well-fitting models to comprise the ensemble.

The resulting ensemble-averaged predictions preserved both the timing and magnitude of the predicted epidemic from the best-fitting model, albeit with wider prediction intervals reflecting structural uncertainties (Supplementary Fig. 10). The estimates of the frequency of infections from each transmission pathway were similar (Supplementary Fig. 11).

**Table 1 | Posterior median and 95% credible intervals of numerically estimated parameters obtained from best-fitting 100 retained simulations (out of 2600 total simulations)**

| Parameter Class | Parameter representation | Description | Posterior median (95% credible interval) | Prior |
|---|---|---|---|---|
| Weekly transmission rate | $\beta_{ij}^{hi}$ | Farm-to-farm (high transmission) | 0.312 (0.023–0.921) | Uniform (0, 1) |
| | $\beta_{ij}^{med}$ | Farm-to-farm (medium transmission) | 0.362 (0.026–0.917) | Uniform (0, 1) |
| | $\beta_{ij}^{low}$ | Farm-to-farm (low transmission) | 0.259 (0.018–0.834) | Uniform (0, 1) |
| | $B_{pj}$ | Patch-to-farm | 3.186 (0.154–5.677) | Uniform (0, 6) |
| | $\beta_{xj}$ | External-to-farm | 0.004 (0–0.01) | Uniform (0, 0.01) |
| | $\beta_{iq}$ | Farm-to-patch | 1.984 (0.08–5.133) | Uniform (0, 6) |
| | $\beta_{pq}$ | Patch-to-patch (overall) | 0.85 (0.018–6.876) | Uniform (0, 8) |
| Relative susceptibility | $\phi_j$ | Of villages during winter slaughter | 0.344 (0.024–0.962) | Uniform (0, 1) |
| | $\phi_q$ | Of patches below forest coverage threshold | 0.029 (0.002–0.668) | Uniform (0, 1) |
| Relative infectivity | $\psi_i$ | Of villages during winter slaughter | 0.51 (0.036–0.893) | Uniform (0, 1) |
| | $\psi_p$ | Of patches below forest coverage threshold | 0.298 (0.008–0.863) | Uniform (0, 1) |

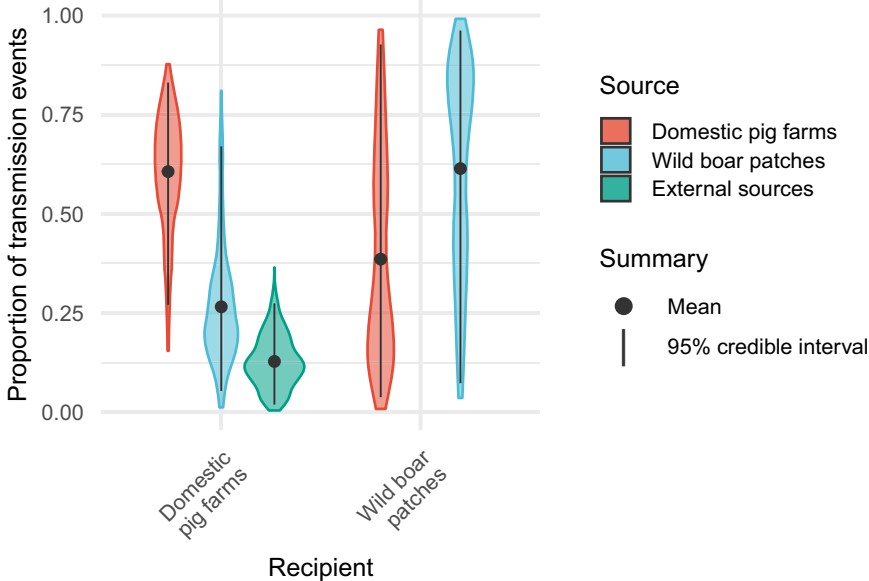

**Fig. 2 | Estimated contribution of transmission pathways to overall epidemic propagation, based on 500 simulations (over 100 particle sets) of the best-fitting model.** For each recipient (x-axis), the color-coded violin plots show the distribution of the proportion of infections attributed to each source. The central dot and bar indicate the posterior mean and 95% credible intervals (CI95). Domestic pig farms were most frequently infected by other domestic pig farms (mean: 60%, CI95: 27–83%), followed by wild boar patches (27%, 5.3–67%) and external sources (13%, 1.9–27%). Conversely, most infections in wild boar patches originated from other wild boar patches (61%, 7.3–96%), with the remainder attributed to domestic pig farms (39%, 3.8–93%).

When applied to the impact of alternative control strategies, performance-weighted ensemble estimates yielded reductions in final epidemic size consistent with those obtained from the single best-fitting model. For preventive and reactive culling among domestic pigs, median reductions of 104 (95% credible interval: −51 to 262) and 62 (−92 to 211) farms were estimated, respectively. Environmental sanitation and improved passive surveillance yielded smaller median reductions in epidemic size by 10 (−125 to 160) and 18 (−125 to 170) farms, respectively. In all cases, the ensemble credible intervals contained zero and were narrower than those seen from the best-fitting model (the latter an expected consequence of averaging across structurally similar models and thereby reducing model-specific variance), indicating that the estimated effect sizes were stable across model structures.

**Sensitivity analysis**

A sensitivity analysis on the surveillance zone multiplier parameter, representing the increase in detection rate within a surveillance zone, revealed minimal effect of this parameter on influencing final epidemic size (Supplementary Table 1).

## Discussion

In this study, we developed a spatially-explicit multi-host mechanistic model of ASF transmission, jointly simulating spread within and between domestic pig and wild boar populations. In contrast to previous ASF models that typically focused on single-host scenarios[20], our framework explicitly integrated stochastic bidirectional interspecific transmission in a setting dominated by low-biosecurity backyard pig farming.

This study and modeling approach advances the field in two key ways. First, it has provided quantitative estimates of interspecific transmission events of a real ASF epidemic; information that is highly beneficial for informing control strategy implementation choices. Second, it demonstrates a flexible, rapidly deployable model framework that can be parameterized with readily available and imperfect

**Table 2 | Effect of alternative control strategies on final epidemic size compared to the baseline scenario**

| Scenario | Host | Baseline median | Scenario median | Median of paired differences | 95% CI of paired differences | P (scenario <baseline) |
|---|---|---|---|---|---|---|
| Environmental sanitation | Domestic pigs | 550.5 | 533.5 | −15 | −664 to 641 | 0.519 |
| | Wild boar | 274.0 | 244.0 | −28 | −357 to 286 | 0.569 |
| Improved passive surveillance | Domestic pigs | 550.5 | 532.5 | −26 | −676 to 631 | 0.528 |
| | Wild boar | 274.0 | 262.0 | −17 | −353 to 313 | 0.538 |
| Preventive culling | Domestic pigs | 550.5 | 444.0 | −97 | −715 to 558 | 0.611 |
| | Wild boar | 274.0 | 263.5 | −17 | −351 to 311 | 0.535 |
| Reactive culling | Domestic pigs | 550.5 | 460.5 | −73 | −706 to 572 | 0.589 |
| | Wild boar | 274.0 | 241.5 | −38 | −364 to 295 | 0.587 |

Baseline and scenario medians report the median final epidemic size across posterior predictive draws. Median of paired differences is not the difference of these two median values, rather the median of the distribution of paired scenario–baseline differences, and is summarized alongside its 95% credible interval (CI). P(scenario <baseline) is the proportion of paired posterior draws in which the final epidemic size in the alternative scenario was smaller than that of the baseline scenario. Scenarios were modeled as modifications to surveillance or culling practices: (1) Environmental sanitation via carcass removal within two weeks of wild boar case detection; (2) Improved passive surveillance of domestic pig farms decreasing the detection delay from three to two weeks; (3) Preventive culling of an entire farm upon detection of the first case; 4) Reactive culling of nearby farms following detection of a wild boar case.

surveillance data, and applied for near-real-time epidemic decision-making. Given that multiple reportable European livestock diseases have a wildlife component[25], an integrated framework represents a needed advancement towards effective, evidence-based management of emerging cross-species livestock diseases.

Our results demonstrated that while domestic pigs and wild boar each sustained their own epidemics, interspecific transmission played a non-negligible role. These estimates were aligned with evidence from South Korea[26], and highlight the importance of developing coordinated interventions that target both host populations. Future decisions around vaccine deployment should also consider this interdependence to maximize effectiveness[27].

In Romania, transmission patterns were shaped by the prevalence of backyard pig farming and imperfect wildlife surveillance, whose intensity varied by county[23]. In our model, each farm unit represented the entire backyard pig population of a village, consistent with the presumed homogeneity from the frequent informal contacts expected between households in a rural village environment. Farm-to-farm transmission, estimated to occur approximately every two to three weeks, could therefore be better interpreted as village-to-village transmission. This rate, slower than the farm-to-farm transmission rate reported for the Russian Federation[28], likely reflects differences between modeled village-level units and actual individual farms, as well as the reduced frequency of movements as they would occur between villages, compared to movements between farms.

Wild boar spread was more rapid than the velocities observed in Poland and the Baltic states[2,3,29,30], although such comparisons are complicated by differences in tessellation definitions and spatial resolution, which have been shown to shape both parameter estimates and downstream inference[31]. The absence of an external force of infection for wild boar in the best-fitting model suggests that key drivers of ASF spread in wild boar—such as local transmission processes and environmental persistence—were adequately captured. The long-range transmission mechanisms that an external force of infection would represent may already be incorporated in our model through allowing for cross-host transmission from domestic pigs, which would implicitly capture the role of human-mediated movements in long-distance spread. This aligns with previous findings that wild boar movement patterns by themselves are poor predictors of ASF dispersal[29], and suggests that human transport, followed by local interspecific transmission, plays a dominant role in sustaining broader-scale epidemics in the wild compartment.

Relying on forest coverage as a habitat suitability proxy rather than density estimates resulted in improved transmission modeling among wild boar, reflecting established ecology[32,33]. The model also captured the reduction in village susceptibility linked to the winter backyard pig slaughter[23,34]. This dynamic corresponded with Romania's December holiday period, during which pig slaughter is widespread and has been seen to influence the transmission of swine-origin diseases. While this seasonal practice appears to have resulted in reductions of ASF spread and subsequent epidemic fade-out—as would be expected to result from a massive reduction in the susceptible population—it has also been associated with increased human trichinellosis outbreaks from pork consumption during Christmas celebrations[35]. In both cases, mass backyard pig slaughter for consumption is a common driver, reinforcing the importance of integrating cultural behaviors into epidemic models.

Our results showed that alternative control strategies could potentially reduce the median epidemic size among domestic pig farms, though Bayesian comparisons revealed only weak evidence that these reductions exceeded what would otherwise be expected by chance. This suggests that the control strategy framework in place during the initial epidemic wave—especially of the epidemic establishment and exponential growth phases that are dominated by stochastic dynamics[36]—was already insufficient to suppress transmission. Simply bolstering existing measures, without reconsidering changes in their scope or design (such as assuming local transmission distance beyond 10 km), would have been unlikely to result in effective control. However, strategies like preventive culling face significant challenges in Romania due to a low social acceptability[37], which limits the feasibility of their implementation despite demonstrated success in other contexts[38].

While our model could mostly replicate spatial and temporal epidemic trends, this occurred under simplified conditions: each village was treated as a single epidemiological unit, thus excluding within-village dynamics and omitting heterogeneity between individual backyard farms. In reality, the magnitude and direction of between-village spread likely reflect differences among village infectiousness profiles, which would be expected to be shaped by the spatial distribution and size of individual holdings within each village[39]. These dynamics could be modeled following identification and geolocation of individual backyard farms; such data is not currently collected, and, if it were, would require considerable effort to compile and structure it for model use.

An additional discrepancy was the sharp, early peak in Tulcea that was not fully captured by our model. This peak was likely a consequence of the stochastic dynamics typical during epidemic onset, whose processes were not considered in our modeling framework. Though surveillance capacity was invariably limited at this early stage, we assumed a constant detection effort that reflected average

surveillance across the modeled period. Rapid, concentrated intensification of surveillance following initial case detection could have produced the observed surge in reported cases that a model with fixed detection probability would not be able to reproduce. Future work accounting for time-varying surveillance efforts (in addition to within-village heterogeneity) may help to address this disparity.

The wide prediction intervals associated with interspecific transmission estimates reflect both epidemiological and model structural uncertainty. In the examined epidemic, summary estimates of interspecific transmission were hampered by sparse surveillance among wild boar, imbalanced surveillance between hosts, and low observed event counts. While ensemble analysis demonstrated that the mean contributions of each transmission route remained stable across alternative model structures, the full scope of plausible transmission frequencies remained wide. This uncertainty likely arises from the limitations of what can be inferred from the available epidemic data through our framework, and supports the importance of improved wildlife surveillance for diseases where non-negligible interspecific transmission events may occur.

Including within-village dynamics could also enable us to account for apparent village re-infections, which were not explicitly modeled, though were observed to affect 10% percent of farms. In part, the observed occurrence of re-infections was due to our blanket definition of recovery (assuming every village was cleared of infection following three weeks without a detected case). However, individual behaviors that would maintain a residual risk of re-infection were suspected to have occurred, such as neighbors to infected farms pre-emptively slaughtering and storing carcasses prior to official intervention, or continuation of the illegal practice of swill feeding[40]. Following the inclusion of these dynamics, re-infection capability could be considered for farms in the model.

Lastly, non-exhaustive wild boar surveillance during the period of epidemic emergence likely resulted in substantial under-detection of cases, particularly in remote or hard-to-access areas like the Danube delta. Future model versions could account for imperfect detection through incorporating a partial observation process that would modulate the probability of case reporting by a given metric (e.g. the phase of the epidemic and recent domestic pig incidence).

Achieving control of an ASF outbreak is a complex task that requires consideration of management impacts across domestic and wild populations, and coordinated efforts from veterinary services, farmers, conservation and hunting groups, law enforcement, and government authorities. In areas dominated by backyard farming, where domestic pig herds and wild boar can each act as reservoirs for the other, control strategies targeting only one host are unlikely to achieve a disease-free state. Without explicit efforts to reduce transmission in both compartments, it may be necessary to adjust epidemic control objectives from eradication to consideration of low levels of circulation in the wild compartment, while supporting sustained biosecurity and surveillance to mitigate transmission back into domestic pig populations. The use of mathematical modeling is already standard practice for decision-making in human public health[41], and with sufficient resource availability, it can become an equally central tool in veterinary public health.

## Methods
### Data and model agents
Six datasets were used to characterize the domestic pig and wild boar populations and outbreak progression. Outbreak records—which included dates, coordinates, host classification (swine or wild boar), and farm type (backyard or commercial)—were retrieved from the WOAH Animal Disease Information System database, covering the period from first detection (10 June 2018) to the end of the first epidemic wave (31 December 2018)[42]. The study region was restricted to the six southeastern counties (Braila, Calarasi, Constanta, Galati,

Ialomita, and Tulcea) where the initial epidemic wave was observed. Village spatial data was retrieved from the Romanian National Agency for Mapping and Real Estate[43]. Locations of industrial production sites were retrieved from data available through the county-level directorates of the National Sanitary Veterinary and Food Safety Authority[44]. Village census data was retrieved from the National Institute of Statistics (INSSE)[45].

Domestic pig farms were categorized into backyard village herds and commercial sites. Though the two classes differ in scale and structure, commercial sites (n = 55 of 1041 total farms) were not considered separately, given both their limited number and their being principally large integrative herds with low connectivity to backyard herds. Conclusions from field investigations recommended that outbreaks be counted in the form of infected and cleared villages[37], resulting in a village being considered as a single large backyard farm and represented through its centroid.

Wild boar habitat patches were characterized through both CORINE Land Cover satellite imagery[46] and ENETWILD wild boar density estimates[47]. Interconnected hexagonal patches, spanning the full study area, were sized to match estimated Romanian wild boar home ranges (25 km$^2$)[32]. For each patch, forest coverage was computed as the proportion of the total area classified as forest, while wild boar density was computed as the mean density across all 4 km$^2$ density grid cells contained within each patch.

Infectious periods were defined through both field expertise and temporal clustering of cases. Sequential cases within a village were considered to be part of the same outbreak if they occurred less than $n$ weeks apart, where $n$ represents the assumed duration of post-detection infectiousness. Expert opinion from field investigations suggested a period of outbreak infectiousness of 2–4 weeks, where cases detected within that period within a village were considered to be part of the same outbreak[23]. Therefore, a village could have multiple outbreaks when gaps between detections exceeded this threshold. To identify an appropriate assumption for infectiousness duration that minimizes spurious re-infections (i.e. those artificially created from definition constraints) while remaining sensitive to true re-infection occurrences, we evaluated a range of candidate durations and applied an elbow method heuristic to select the value prior to diminishing returns of increasing infectiousness duration (Supplementary Fig. 12). Here, a value of three weeks was identified and subsequently used to compute the individual infectious periods for each village. Wild boar patches were assumed to be infectious for six weeks following case detection[48].

To assess the potential influence of within-village heterogeneity, we explored the association between village population size and infectious period duration, considering village population a proxy for family counts and thus farm counts. Only a weak relationship was identified (Spearman's = 0.10), suggesting that village size did not explain the variation observed in the infectious period durations. Given the absence of additional within-village farm data, within-village dynamics were not explicitly modeled.

In the model, each agent is either a domestic pig farm (denoted $i$ if infectious or $j$ if susceptible) or a wild boar patch (denoted $p$ if infectious or $q$ if susceptible). Units are generically referred to as source unit $a \in \{i, p\}$ and target unit $b \in \{j, q\}$, depending on the infection state and host class.

### Infection process
The infection state process for each agent followed a modified SIR (susceptible, infectious, recovered) framework. To reflect the surveillance process, the infectious state was partitioned into undetected and detected compartments. Domestic pig farms transited through susceptible, infectious, detected (and still infectious), and recovered states, while wild boar patches transited through susceptible, infectious, and detected states. Latency was not considered as the model

used weekly timesteps and the mean incubation period for the ASFV genotype II strain-of-interest was less than one week[49–51]. Re-susceptibility was not considered, as re-infection events were uncommon (10% of farms), and accounting for this was unsupported by the data.

Each susceptible unit $b \in \{j, q\}$ received infectious pressure from internal sources $a \in \{i, p\}$ plus external sources (representing long-distance movements or cross-border introductions), yielding a total of six forces of infection. The total force of infection on susceptible unit $b$ at time $t$, given by $\lambda_b(t)$, was computed as:

$$\lambda_b(t) = \phi_b \left( \sum_a \lambda_{ab}(t) + \lambda_{xb}(t) \right) \quad (1)$$

Where $\phi_b$ represented the relative susceptibility of target unit $b$, which was given by the assumed timing of the winter holiday slaughter period for farms and by habitat suitability for patches; $\lambda_{ab}(t)$ represents the total force of infection from all units $a$ onto unit $b$ at time $t$; and $\lambda_{xb}(t)$ represents the total force of infection from external sources onto unit $b$ at time $t$. The individual infectious pressure from each source of infection $a$ onto susceptible unit $b$ at time $(t)$ was computed as:

$$\lambda_{ab}(t) = \mathbb{1}_{a_{\text{inf}}}(t) \cdot \mathbb{1}_{b_{\text{sus}}}(t) \cdot \mathbb{1}_{ab_{\text{contact}}} \cdot \psi_a \cdot \beta_{ab} \quad (2)$$

Where indicators $\mathbb{1}_{a_{\text{inf}}}(t)$, $\mathbb{1}_{b_{\text{sus}}}(t)$, and $\mathbb{1}_{ab_{\text{contact}}}$ indicated if unit $a$ was infectious at time $t$, unit $b$ was susceptible at time $t$, and units $a$ and $b$ were in contact, respectively; $\psi_a$ represented the relative infectivity of source unit $a$, modeled in the same manner as $\phi_b$; and $\beta_{ab}$ represented the weekly transmission rate from source unit $a$ onto target unit $b$ for unit class pair $ab$. For farms, both the relative infectivity and susceptibility were adjusted to reflect the shift in transmission dynamics that would be presumed to occur from a mass cull. This parameter took effect in model week 24, corresponding to the start of December, to capture both the official slaughter date (December 20) and account for preceding preparatory activities.

The external force of infection from external sources $x$ upon unit $b$ was computed as:

$$\lambda_{xb}(t) = \beta_{xb}(t) \quad (3)$$

where $\beta_{xb}(t)$ represented the weekly transmission rate from external sources onto unit $b$ at time $t$. The external force of infection was applied dynamically per county, determined by the estimated first week of infection in a given county and adjusted by the assumed detection delay for the host class.

Transitions among infection states were simulated using a classical τ-leap approximation[52], in which discrete-time transition probabilities were derived from exponential waiting times and applied at fixed time steps. For susceptible unit $b \in \{j, q\}$, infection occurred with the probability:

$$p_{\text{inf}, b}(t) = 1 - \exp(-\lambda_b(t) \cdot dt) \quad (4)$$

Where $\lambda_b(t)$ was the total force of infection on unit $b$ at time $t$ (as defined in Eq. 1), and $dt$ represents the fixed simulation time step (set at 0.1, one-tenth of a week). To account for multiple potential sources of infection, a Bernoulli trial was independently evaluated for each transmission route at each time step. If infection of the target unit was successful, the responsible transmission route was sampled proportionally to the normalized probabilities of successful transmission. Upon infection, the unit entered the infectious-undetected state.

Transition from the infectious-undetected to infectious-detected state was governed by the detection rate $\sigma$, and assumed to follow an exponential distribution. The corresponding probability of passive

detection at each τ-leap time step for unit $a$, given its unit class, was:

$$p_{\text{det, passive}} = 1 - \exp(-\sigma_a \cdot dt) \quad (5)$$

For farms under active surveillance, defined as those within a 10 km radius of a detected farm as per national regulation[24], the detection probability was increased by a multiplicative parameter α. The resulting probability of active detection at each time step was:

$$p_{\text{det, active}} = 1 - \exp(-\sigma_a^{\text{farm}} \cdot \alpha \cdot dt) \quad (6)$$

Assuming that all cases are eventually detected, transition to the recovered state for detected units was governed by the recovery rate $\gamma$, specific for unit class and also assumed to follow an exponential distribution, and given by the probability:

$$p_{\text{rec}} = 1 - \exp(-\gamma_a \cdot dt) \quad (7)$$

No recovery was assumed for wild boar patches, based on evidence regarding carcass persistence, imperfect carcass detection, and environmental contamination durations[53,54].

## Model structure and variant formulations

Variations of multiple key structural parameters were tested to identify which parameter formulation best-captured the observed epidemic, including contact structure, transmission pathway formulation, and transmission rate formulation (Supplementary Table 2).

Contact from farms to patches was defined as either zero-order (contacting only to the patch containing the farm) or first-order (contacting both the patch containing the farm and its immediate neighbors). Similarly, contact from patches to farms was defined as either zero-order (contacting farms within the patch only) or first-order (including farms in adjacent patches). Contact among farms remained fixed at 10 km, and patch-to-patch contact was limited to first-order adjacency.

Each transmission pathway (*ij*, *pj*, *iq*) was evaluated under both frequency-dependent and density-dependent assumptions, with the exception of patch-to-patch (*pq*) transmission, which was always frequency-dependent due to the near-uniform six-neighbor configuration of the hexagonal patch network.

Transmission rate structures were formulated to reflect the spatial heterogeneity seen in the observed county-level trajectories. For farm-to-farm transmission ($\beta_{ij}$), both a uniform transmission rate across all counties and a transmission rate stratified into high, medium, and low tiers were used. The tiers corresponded to the total number of infected farms in the observed data, with Tulcea ($n = 106$) and Braila ($n = 87$) classified as high-transmission, Constanta ($n = 51$) and Ialomita ($n = 42$) as medium, and Calarasi ($n = 20$) and Galati ($n = 8$) as low.

For patch-to-patch transmission ($\beta_{pq}$), models took either a uniform transmission rate across all counties or followed a two-tier system where counties with a high incidence among wild boar (Tulcea, $n = 37$; Ialomita, $n = 11$) were assigned a separate rate.

Lastly, the transmission rate from external sources onto wild boar patches ($\beta_{xq}$) was either active, where its application varied temporally and was applied to patches within counties aligned to start with the estimated date of first infection, or inactive, where its effect was removed from the model.

These combinations resulted in 256 total models that contained between 9 and 13 estimated parameters, depending on whether specific transmission rates were stratified by county-level incidence and whether an explicit external force of infection upon wild boar was included.

All models used the same state transition framework and initial conditions.

## Model initialization

Simulations were initialized by seeding infections in the two wild boar habitat patches that contained the earliest confirmed wild boar cases, which were also the two patches that contained the first domestic pig farm cases. These detections, having occurred on 12 and 16 June 2018 in Tulcea county, were backdated by 6 weeks to account for the estimated detection delay for wild boar carcasses[55]. The simulation, therefore, started in early May 2018.

## Model calibration

To select the most appropriate habitat suitability proxy for wild boar transmission risk, a ROC curve for predicting the occurrence of at least one case in a patch was constructed for both forest coverage and host density estimates.

Following the determination of the metric for habitat suitability, the array of candidate models reflecting all combinations of parameter structures was calibrated to identify which parameter combinations and structural assumptions best replicated the observed epidemic dynamics. Each model estimated the posterior distributions for a set of unobservable transmission parameters, consisting of transmission rates $\left(\beta_{ij}\,\beta_{pj}\,\beta_{xj}\,\beta_{iq}\,\beta_{pq}\,\beta_{xq}\right)$ and relative modifiers for host infectivity $\left(\psi_i\psi_p,\right)$ and susceptibility $\left(\phi_j\phi_q\right)$. Contingent on model structure, certain parameters were stratified by county incidence or omitted altogether. Detection and recovery rates were informed from the literature and observed data. Detection in farms was estimated at 1/3 weeks$^{-1}$ based on outbreak investigations in the Russian Federation[56], while detection in patches was set at 1/6 weeks$^{-1}$ based on estimates for a large wild boar population[55]. Recovery rates for farms were computed empirically from the observed epidemic data and defined per county, per time period (split between the first and second halves of the model period), while patches (if recoverable, as in the alternate scenario) were assigned a fixed rate of 1/3 weeks$^{-1}$.

Numerical parameterization occurred through approximate Bayesian methods using a sequential Monte Carlo algorithm (ABC-SMC), implemented via the adaptive population Monte Carlo (APMC) methodology[57].

At each calibration step, 200 parameter sets ("particles") were drawn from uniform prior distributions. Epidemic trajectories were generated per particle for each host population and summarized as the incidence counts per host, per county, per week—for a total of 432 summary statistics. To avoid over-weighting statistics with high variance, the contribution of each statistic to the overall distance was normalized by its variance, ensuring compatibility among the different magnitudes of the individual summary statistics. The final distance between the simulated and observed data for the particle was then computed as the maximum absolute distance across all summary statistics.

Particles having simulated distances below a dynamically-updated tolerance threshold were retained, with the proportion of particles accepted at each step used to determine convergence. Retained particles were assigned weights based on their proximity to the observed data and perturbed to augment exploration of the parameter space, and a new set of particles was generated. At each successive generation, the tolerance threshold was reduced, forcing the algorithm to generate simulations increasingly closer to the observed summary statistics. This process continued until the particle acceptance rate fell below 5%, the recommended termination threshold, where limited improvement in convergence would be obtained from further iteration.

## Model selection and structural evaluation

All models were ranked by their final Euclidean distance to the observed summary statistics, and a filter was applied to retain only those that succeeded in capturing the observed epidemic trajectory in at least 4 counties. This ensured that models that closely fit the overall summary statistics but failed to reproduce important localized peaks in incidence (a product of the sparsity of the observed incidence in some counties) were excluded. The retained model with the lowest summary statistic distance was then selected as the best-fitting model.

To evaluate the contribution of each structural parameter to overall model fit, a random forest regression model was used with final distance as the outcome variable and the structural parameters as predictors. Corresponding variable importance scores were used to quantify the influence of each parameter. To determine which variables associated with each structural parameter yielded better-fitting models, empirical cumulative distribution functions of the final distance across the values of each structural parameter were visually assessed.

## Baseline and alternative management scenarios

To evaluate the impact of enhanced control strategies on epidemic outcomes, both a baseline simulation and four alternative management scenarios—that targeted either improved detection or reduced transmission—were explored through their effect on final epidemic size. For each scenario, five replicates of each conserved particle were run, resulting in a total of 500 simulations per scenario. Results were summarized by the median and 95% credible intervals of the final epidemic size.

Improved passive surveillance among farms was simulated by reducing the average duration of undetected circulation from three to two weeks. Reactive culling was implemented through shortening the infectious period of detected farms to one week, reflecting immediate depopulation upon detection. Preventive culling in response to nearby wild boar case detection was simulated through transitioning all farms within a detected patch to the recovered state upon detection. Environmental sanitation, reflecting clearance of wild boar carcasses, was simulated through patches transitioning to the recovered state following a two-week period of circulation—reflecting the time needed to disinfect the area—rather than remaining infectious indefinitely.

## Performance-weighted ensemble modeling

To account for structural uncertainty and to avoid reliance on a single best-fitting model, performance-weighted ensembles from the top model variants (defined as those that successfully captured the observed epidemic trajectory in at least four counties, clustered at the lowest distance, and provided a satisfactory visual epidemic fit) were constructed via a model-averaging approach[58]. Model weights were computed as the normalized proportion of the predictive error of each model, with the final Euclidean distance between simulated and observed summary statistics serving as the performance marker. This allowed models with a better fit to have a proportionally greater influence on ensemble estimates.

The derived weights were applied to the summary outputs of each model (weekly incidence medians and credible interval bounds, proportion of infections attributable to each transmission pathway, and distributions of final epidemic size under baseline and alternative scenarios), and ensemble estimates accounting for the relative influence of each model were obtained.

Data processing, model implementation, parameterization and analysis were performed in R statistical software version 4.3.3 "Angel Food Cake"[59]. The sequential Monte Carlo algorithm for model calibration was implemented via the *EasyABC* package[60], and random forest feature selection occurred via the *ranger* package[61]. Data manipulation and visualization were performed via the *tidyverse* suite[62].

## Reporting summary

Further information on research design is available in the Nature Portfolio Reporting Summary linked to this article.

## Data availability

Data are available at: https://gitlab.envt.fr/epidesa/asf-multihost-romania/.

## Code availability

Full scripts are available at: https://gitlab.envt.fr/epidesa/asf-multihost-romania/.

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

## Acknowledgements

The authors would like to thank Inaporc and Agreenium for funding this research; the Romanian veterinary authorities for their field expertise; the INRAE MIGALE bioinformatics facility[63] for providing computing and storage resources; and Pierre Bourcier, scientific and medical illustrator at globulocreation.com, for graphical abstract illustration.

## Author contributions

B.H., T.V., N.R., and M.A. designed research; B.H. performed research; B.H., T.V., N.R., and M.A. analyzed data; B.H. wrote the paper, and T.V, N.R., C.M., and M.A. reviewed the manuscript.

## Competing interests

The authors declare no competing interests.
