## [Transparent Peer Review file · Nature Communications]

A multi-host mechanistic model of African swine fever emergence and control in Romania

Corresponding Author: Dr Brandon Hayes

Version 0:

Reviewer comments:

Reviewer #1

(Remarks to the Author)

I am writing upon reviewing the article entitled 'A multi-host mechanistic model of African swine fever emergence and control in Romania' by Hayes et al.. Clearly a lot of technical work has gone into this study, however I found that the article and supplied code obstructed my ability to assess (i) what the noteworthy results were, (ii) if the work is of significance to the field and (iii) whether the work support the conclusions and claims of the study. I further had trouble running the code on the online repository (as detailed in 'Remarks on code availability') and did not find the methodologies detailed enough for a reader to be able to reproduce the study. Substantial alterations to the manuscript would need to be made to improve it. In the future, I would also request that the authors add line numbering to their manuscript to make it easier for reviewers to give their comments. Below I have outlined the improvements that could be made throughout the manuscript.

INTRODUCTION

1. The introduction states that 'interrupting disease transmission is reliant on control strategies targeting these pathways' and later, 'evaluating potential outcomes from alternative control strategies'. Are the alternative control strategies that are evaluated in the model different from the current controls? It would be useful for the reader if more detail is included on what the current controls are, and how the models evaluate an 'alternative'.

RESULTS

1. The methodologies are understandably important, and owed to the format of the journal (i.e. Methods at the end), it would be beneficial if the results had a much gentler first paragraph covering what does the model attempt to capture, the differences between the fitted models, and what kind of data was it fitted to (away from technical jargon).
2. In the first paragraph of results, you should define 'MSE' given that the Methods come later.
3. Was a particular method used to determine that the best fitting model's prediction intervals 'satisfactorily' captured the weekly incidence for each host?
4. The work that has gone into fitting all 24 models and estimating uncertainty is substantial. To that end, in the last 2 sentences at the end of paragraph 3 of the Results, can the uncertainty of the time estimates also be supplied in addition to the median?
5. Related to the above point, can you please maintain consistency between the intervals presented: throughout there is a mixture between only a point estimate, the inter-quartile range, a 95% equal-tailed interval, and a 95% highest-density interval.
6. Within the results section, can you explicitly add into each subsection how many parameter samples and stochastic simulations were used to generate of the summary statistics?

INTERPRETATION

The discussion is far too long which makes it very challenging to consolidate the study. Mainly, the length obstructs the reader in determining what the noteworthy results are, and if they are of significance to the field.

1. Please collapse the discussion down significantly.
2. The paragraphs in the discussion are not consistent: start with what your findings were for each section, and then compare these to other studies.
3. Please adjust the final sentence of the first paragraph in the Discussion reading 'The final result was a model where the overall trends observed in the epidemic emergence period were able to be replicated through stochastic simulation.' It needs to be made more clear

that only the 'the spatial distribution of the epidemic is further replicated in the majority of areas', and not all areas.

4. There are spikes in domestic pig cases in Braila and Tulcea not explained by the model (Results paragraph 2 & figure S2). What factors not included in the model could explain the inability to capture the spike in weekly incidence in Braila and Tulcea?
5. It would be interesting to see how estimated model parameters differ depending on the twenty-four model designs.

METHODOLOGY

The study would not be able to be replicated given the level of detail as it stands in the Methods and does not fulfil the standard as expected in the field. Below are some points that could help improve this section:

1. Please consider rewording the first few sentences of the model agents section. Referring to the models in domestic pig herds and wild boar cells before they are defined is confusing.
2. Please be more explicit in defining the compartmentalisation of the model.
3. Given the forces of infection, the full set of transitions needs to be defined mathematically.
4. The tau-leap algorithm needs to be more explicitly defined as multiple methods are accepted/discussed in practice, not just the classical one.
5. The reference (10) also does not refer the appropriate section of the book.
6. Please avoid the explicit use of '*' to indicate scalar multiplication.
7. Why was the simulation initialised 8 weeks prior to the detection of the first case?
8. All 24 models need to be explicitly described, in particular how it would explicitly change the equations of the model. A table would be very helpful here to allow the reader to quickly compare the models.
9. It's unclear how many parameters there actually are in the model. The Methods states 15, Table 1 indicates 12 and a further 24 depending on the village and quarter. Do the number of parameters also change with the model design?
10. In the ABC, how were the 56 summary statistics used coerced into a single value describing distance between the model and data?
11. Why was MSE used to determine model selection, and why was it done on a scale completely different to the summary statistics used to fit the models?
12. It's not clear how control outcomes were compared with the baseline outcomes as there are several options particularly given it's a stochastic model.
13. This applies to summary metrics as well as the model fitting, given that you have a stochastic model, how many realisations of the model were there per parameter set?

FIGURES AND TABLES

1. Please add a legend to Figure 1.
2. Figure 1 is not convincing that the model can replicate the observed dynamics of domestic pigs. It is unclear if this is because of the choice of visualisation, i.e. the summary metrics per time unit aggregate any model-specific noise leading to a non-noisy median, or if the model itself is unable to produce generally noisy trajectories. Not only does Figure S2 more appropriately summarise model performance, it would be good to also show:
 - a. model predictions against the data at the level of summary statistics (i.e. quarterly incidence per county and per entire study region for domestic pig units and wild boar cells).
 - b. a typical set of trajectories with a low number of replicates.
3. Please add a legend to Figure 2 and 4 to define the points and error bars (i.e. median and IQR).
4. Table 2 can you please put that these are percentages in the table.
5. An expanded table (as in Figure S1) is needed to describe all 24 models.
6. Supplementary figures showing the model predictions against the data for all other 23 models would also be useful to determine how valuable the model design is in qualitatively capturing the observed dynamics.
7. Table 1 needs a samples size for the 95% credible interval.

TYPOS

page 2 'one of the highest consequence diseases': should this be 'one of the high-consequence diseases' since this refers to a class of diseases.

page 3 'We estimate that a susceptible village was infected': should this be 'We estimated that a susceptible village was infected'.

(Remarks on code availability)

Unfortunately, not all required content was marked as complete on the Code and Software Submission Checklist. As a result, it was challenging to spot-check the supplied R scripts and verify some of the codes outcomes. I would suggest getting a colleague to try this software on a brand new machine to test its portability and if the information supplied in the code repository is sufficient.

Below I have outlined some of the features that would be helpful to include in the README file (in addition to features unchecked in the Code and Software Submission Checklist) and stated the issues encountered while running the code in order to help the authors' address these issues. An explicit demo file would be extremely useful to satisfy the execution criteria of the Code and Software Submission Checklist.

ADDITIONS FOR THE README.TXT

1. The R version and all package versions that were used to generate the results should be included in at least the main README.txt file, or ideally both README.txt files in each subdirectory '/asf-rom-build' and '/asf-rom', as from sessionInfo(). For example:

R version 4.3.2 (2023-10-31 ucrt)
Platform: x86_64-w64-mingw32/x64 (64-bit)
Running under: Windows 11 x64 (build 22631)

attached base packages:

[1] parallel stats graphics grDevices utils datasets methods base

other attached packages:

[1] doParallel_1.0.17 iterators_1.0.14 foreach_1.5.2 stringi_1.8.2 janitor_2.2.0
[6] sf_1.0-15 lubridate_1.9.3 forcats_1.0.0 stringr_1.5.1 dplyr_1.1.4
[11] purrr_1.0.2 readr_2.1.4 tidyr_1.3.0 tibble_3.2.1 ggplot2_3.4.4
[16] tidyverse_2.0.0 plyr_1.8.9

loaded via a namespace (and not attached):

[1] dotCall64_1.1-1 gtable_0.3.4 spam_2.10-0 raster_3.6-26 htmlwidgets_1.6.4
[6] lattice_0.22-5 tzdb_0.4.0 vctrs_0.6.5 tools_4.3.2 crosstalk_1.2.1
[11] generics_0.1.3 proxy_0.4-27 fansi_1.0.6 pkgconfig_2.0.3 KernSmooth_2.23-22
[16] RColorBrewer_1.1-3 leaflet_2.2.1 lifecycle_1.0.4 compiler_4.3.2 fields_15.2
[21] rgeos_0.6-4 munsell_0.5.0 terra_1.7-65 codetools_0.2-19 leafsync_0.1.0
[26] snakecase_0.11.1 stars_0.6-4 htmltools_0.5.7 maps_3.4.2 class_7.3-22
[31] pillar_1.9.0 classInt_0.4-10 lwgeom_0.2-13 abind_1.4-5 tidyselect_1.2.0
[36] digest_0.6.33 fastmap_1.1.1 grid_4.3.2 colorspace_2.1-0 cli_3.6.2
[41] magrittr_2.0.3 base64enc_0.1-3 dichromat_2.0-0.1 XML_3.99-0.16 utf8_1.2.4
[46] leafem_0.2.3 e1071_1.7-14 withr_2.5.2 scales_1.3.0 2.1-2
[51] timechange_0.2.0 RANN_2.6.1 hms_1.1.3 png_0.1-8 tmaptools_3.1-1
[56] tmap_3.3-4 viridisLite_0.4.2 rlang_1.1.2 Rcpp_1.0.11 glue_1.6.2
[61] DBI_1.1.3 rstudioapi_0.15.0 R6_2.5.1 units_0.8-5

2. Typical execution time for one of the shell scripts in '/asf-rom' to fully execute.

3. Note that one package (rgeos) has been archived on CRAN as since development, but can still be downloaded and installed as of writing.

4. Line 102 of '/asf-rom/README.txt' should read 'Bash script "submit-run-par-test-*.sh"' not "'submit-run-par-est-*.sh"'?

5. Please split up SCRIPT FLOW section of '/asf-rom/README.txt' into (i) the commands the user needs to execute to replicate the study, and (ii) what each script does.

EXECUTION ERRORS IN CODE

1. Test of '/asf-rom-build/01-init_data.R' unsuccessful:

- After changing all hard-coded working directories in each file, this script initially this crashed my machine.
- Upon removing parallelism (please don't hard-code #nodes to use in a parallel loop in the future), I encountered this error which I assume comes from f-bin_clc.R#21:
Error in { :
task 1 failed - "cannot derive coordinates from non-numeric matrix"

2. Test of '/asf-rom/bash/submit-run-par-est-412.sh' unsuccessful:

- After changing working directories etc., the script failed on line 96 with
Error in SimulateModel(l0, pars = priors, fixed.pars, herdDist, unitData, :
argument "dt" is missing, with no default
Called from: SimulateModel(l0, pars = priors, fixed.pars, herdDist, unitData,
dt, maxDist, trans.dist.function = trans.dist.function, dpdp.trans.mode = dpdp.trans.mode,
wbdp.trans.mode = wbdp.trans.mode)
 - With some quick debugging, I can see that dt is defined in the global environment from the call to init.model, but cannot be seen within the full traceback of the error:
Error in SimulateModel(l0, pars = priors, fixed.pars, herdDist, unitData, :
argument "dt" is missing, with no default
12. SimulateModel(l0, pars = priors, fixed.pars, herdDist, unitData, dt, maxDist, trans.dist.function = trans.dist.function, dpdp.trans.mode = dpdp.trans.mode, wbdp.trans.mode = wbdp.trans.mode) at function-par-est.R#36
- old_model(param_with_constants)
 - model(param)
 - .ABC_rejection_lhs(model, prior, prior_test, nb_simul, use_seed, seed_count)
 - .ABC_Lenormand(model, prior, prior_test, nb_simul, summary_stat_target, use_seed, dist_weights = dist_weights, verbose, ...)
 - .ABC_sequential(method, model, prior, prior_test, nb_simul, summary_stat_target, use_seed, verbose, dist_weights = dist_weights, ...)
 - .ABC_sequential(method = "Lenormand", model = model, prior = priors,

```
nb_simul = n.sim, summary_stat_target = summary_stat_target,  
p_acc_min = p.acc.min, verbose = TRUE) at function-par-est.R#60  
5. par.est(sub.id, n.sim, p.acc.min, fixed.pars, l0, n.breaks, sum.stats.obs,  
sum.stats.target = sum.stats.target, trans.dist.function = trans.dist.function,  
dmdp.trans.mode = dmdp.trans.mode, wbdp.trans.mode = wbdp.trans.mode) at run-par-est.R#95  
4. eval(ei, envir)  
3. eval(ei, envir)  
2. withVisible(eval(ei, envir))  
1. source("asf-rom/scripts/run-par-est.R")
```

3. Test of '/asf-rom/bash/submit-run-par-test-408.sh' unsuccessful:
a. The same error occurred as above.

Reviewer #2

(Remarks to the Author)

Overall I found the model to be a pleasant and approachable read, and the authors did a decent job describing the model and its results, allowing for the difficulty in describing individual based models, and the journal's "results first" style.

I have two major points of concern:

1) The combination of the model not being able to model individual herds within a village, as well as the reports discussed by the author of various ways to avoid or evade culling, makes me wonder if the model is overestimating the effectiveness of culling. Some discussion of the sensitivity of the model to incomplete culls is probably not uncalled for.

2) Some of the parameters in the supplement appear to be practically if not actually non-identifiable. This should be addressed in the body of the paper itself, as this is of major concern.

(Remarks on code availability)

Code appears well documented. Libraries are a little bit cumbersome as tidyverse is installed in its entirety. Code in 01-init_data.R did not run, as the rgeos library appears to be no longer available.

Reviewer #3

(Remarks to the Author)

In the manuscript entitled "A multi-host mechanistic model of African swine fever emergence and control in Romania", Hayes and collaborator develop a model to assess the variables determining the extension, duration and number of swine affected by an ASF outbreak in Romania, taking into account the specificities of pig production and the wild boar-domestic pig interface in this country. The authors have gathered a reasonable dataset, undertaken a considerable amount of work, and obtained some nice results, but from my point of view they miss to give the article a wider scope, refine their methodology, and synthesize and effectively communicate their most relevant results in a hierarchical and straightforward way. Additionally, the manuscript has a non-negligible number of formal issues that should be addressed. Although the format requirements for Nature communications are rather relaxed, the whole review process would have been much easier and simpler if the manuscript was line-numbered. English needs style and grammar revision, since part of the length of the manuscript can be attributed to not-so-straightforward language and writing, apart from other errors (see specific comments).

While they achieve a relatively acceptable explanation of one country-case and provide some relevant data regarding the percentage of infections in each compartment (i.e., domestic pigs and wild boar) originating from cross-transmissions (which is relevant not only for modelling, but also for biosecurity and management), the Discussion section (and the Policy impact and real-time support within it) fails to communicate an expected impact of the study beyond the adjustment of the model to the description of this one country-case. Most of the Discussion section is focused on justifying the results and comparing them with previous data of other country-cases (see specific comments below), which does not provide a significant advance in knowledge but adds to previously existing literature. Instead, the authors could extend on the applicability and advantages provided by the model.

Other major flaws raising concern include methodological aspects (see specific comments below), with limitations both in the assessment of domestic pig farm infection and, particularly, in the indirect estimation of wild boar population. Refining these two aspects would probably produce more accurate results, as would do extending their modelling period to a complete natural year.

SPECIFIC COMMENTS

Methodology: some criticism could be made to the design and construction of the variables, i.e., assuming villages as a single epidemiological unit; giving the same relevance to all the villages and industrial farming units without considering relevant factors for ASF transmission such as biosecurity and number of pigs per village/industrial farm; ... for example, the number of pigs per intensive farms could be included in the density-dependence of transmission within the model, although admittedly, since intensive breeding pig farms are relatively scarce and supposedly more biosafe than backyard pigs, this would probably not affect significantly the outcome. However, any other option could also be criticized, so the decisions of

the authors can be also valid. The main concern remains about the depth of the information used to construct the model, since more detailed quantitative and qualitative information for the variables used would probably have refined the output, without need to add further variables. The categorical proxy for estimating wild boar population abundance (i.e., above of below 15% of forest over) could be improved, since finer and more detailed data on wild boar population estimates can be achieved through existent models previously developed for Europe, including Romania (see, for example, ENETWILD consortium et al. 2019, 2021). Such models are also available to estimate the wild boar-domestic livestock (including swine) interface (ENETWILD consortium et al. 2020).

The fact that “all models that used density-dependent distance kernels or frequency-dependent step functions performed better than those that used a density-dependent step function” (page 3) further supports the idea of more complete, precise, and detailed data feeding to the model contributing to improve the accuracy and reliability of the output.

- ENETWILD consortium, Acevedo P, Croft S, Smith G, Blanco-Aguilar JA, Fernández-López J, Scandura M, Apollonio M, Ferroglio E, Keuling O, Sange M, Zanet S, Brivio F, Podgorski T, Petrovic K, Soriguer R and Vicente J, 2019. ENETwild modelling of wild boar distribution and abundance: update of occurrence and hunting data-based models. EFSA supporting publication 2019:EN-1674. 29 pp. doi:10.2903/sp.efsa.2019.EN-1674

- ENETWILD consortium, Fernández-López J, Acevedo P, Blanco-Aguilar JA, Vicente J, 2020. Analysis of wild boar-domestic pig interface in Europe: preliminary analysis. EFSA supporting publication 2020:EN-1834. 45 pp. doi:10.2903/sp.efsa.2020.EN-1834

- ENETWILD consortium, Illanas S, Croft S, Smith GC, Fernández-López J, Vicente J, Blanco-Aguilar JA, Pascual-Rico R, Scandura M, Apollonio M, Ferroglio E, Keuling O, Zanet S, Brivio F, Podgorski T, Plis K, Soriguer RC and Acevedo P, 2021. Update of hunting yield-based data models for wild boar and first models based on occurrence for wild ruminants at European scale. EFSA Supporting Publication 2021:EN-6825. 30pp. doi:10.2903/sp.efsa.2021.EN-6825

Page 5: “Research into the social dynamics around pig farming during this time period would help to further clarify the accuracy of this fitted value. As the subsequent epidemic wave in 2019 was initiated at a similar period, it is possible that this temporal consistency is associated with this annual social dynamic, though further investigation is required to confirm this hypothesis”. I have been wondering all throughout the manuscript why the authors selected only a half-year period instead of a year-round period, as most simulation studies do, in order to capture all the seasonality and not just the December slaughter one. Since backyard pig rearing takes place mostly seasonally from July to December (something the reader learns only in the results section), developing a whole year-round model makes perfect sense to capture the epidemiological dynamics of ASF in Romania.

The authors also dismiss in their model the probability of reinfection in villages. I assume that this would complicate both modelling and output interpretation, but a 13% ratio is not negligible.

More importantly, the authors neither model external sources of infection for wild boar, thus yielding three infection rate parameters for domestic pigs but only two for wild boar. Nevertheless, there is abundant literature demonstrating the RELEVANCE of external sources of wild boar infection for the onset of the epidemics. The cases in Romania (as in Poland, Czech Republic, Germany, Belgium, and Italy) started with the detection of wild boars dead from ASF, therefore originating from such “additional sources of infection not otherwise captured” (page 3). The relevance of such infection sources cannot therefore be neglected.

Maybe I have missed it, but the authors mention that the factor regulating the infectivity of domestic pig herds was seasonality, and I have not found in the Materials and methods section a description with the adequate reference for the seasons with higher and lower ASF infection susceptibility. In the results (pages 3 and 5, respectively), the reader learns that it is in winter (“winter holiday slaughter period”) and specifically in December (“December festivals”), but the precise dates, the parameters, and the reasons and references supporting the higher risk are still missing and should be further and better explained in the Materials and methods section.

“Transmission between herds was further informed by a step function bound at 20 km for the distance between infectious herd i and susceptible herd j ($d!_{i,j}$)” Why 20 km? ASF transmission distance among herds, in absence of pig movements between farms, is supposed to be lower, since ASF is not air-transmitted as CSF but transmitted by close contact, isn't it? Actually, active ASF surveillance after an outbreak in a farm extends over a 10 km radius, as acknowledged by the authors in the Materials and methods section. How is then supposed an infected herd be infective for another herd at 20 km? At the end of page 7 the reader can find an explanation for the 20-km range (“to capture non-reported pig movements between villages which principally occur at a small scale, our maximum distance was increased accordingly”), but for this assumption to be valid the authors should include it in the Materials and methods section with a reliable reference. Otherwise, as aforementioned, the selection of 20 km, even if intended to account for inter-village commercial pig movements, is random and a non-scientific assumption.

In fact, there is a whole paragraph in the Results section (page 3: “Though the distance kernel (δ) for transmission between domestic pig units was estimated as well, the best fitting model utilized a step function (with the force of infection constant at distances shorter than 20 km and null otherwise) to scale the force of infection between domestic units with increasing distance. We estimate that a susceptible village was infected by an infectious village a median of every three weeks, while a susceptible wild boar habitat cell was infected by a neighbouring infectious cell every four weeks. The transmission rate from an infected cell to village herds was estimated at approximately two cases every three weeks, and the transmission rate from village herds to the local wild boar environment was estimated at approximately one case per week”) that is rather Materials and methods. Here the authors are not communicating their results but describing the parameters they chose for

the model. This part should not only be transferred to the Materials and methods section, but also adequately referenced (otherwise all the transmission rates selected appear random). Probably after searching and adding the adequate references the authors will consider re-doing some analyses. For example, as aforementioned, a transmission from village-to-village constant below 20 km and null over 20 km does not look adequately and finely scaled to me. Instead of choosing invariable values for these (and other) parameters to be included in the model, I suggest the authors to carry out sensitivity tests to verify the best-fitting values for the parameters they use to construct the model. This is probably another way to potentially improve the accuracy and predictive value of their model.

The authors justify that their best-fit models (figures 1 and S2) capture the actual epidemic trends are included within the 99% interval both for domestic pig units and wild boar cells, either considered altogether or separately by county. However, they fall to capture the minor detail of the epidemic waves, particularly after they reach the peak and the epidemic fades away or turns into endemicity and particularly for the specific county cases.

The 99%, credible interval of the scenarios provided by the model are “generous” in forecasting a high spread of the disease, thus it is not surprise that the real epidemic waves fall included within the forecasted prevalence. However, looking at the detailed trend, the correspondence is not so accurate. This inaccuracy improves when the counties are pooled for the whole of the study area (figure 1), but I guess it could be still more refined if increasing the detail of the input variables (quantitative numbers for pig breeding sites, progressive transmission distance between pig farms up to 20 km, finer wild boar population estimations, ... The low effect of environmental sanitation (i.e., removal of ASF-dead wild boar carcasses) proposed by the model in the alternative management scenarios is another evidence of the space for improvement and refinement of the model, since in the real (not simulated) world this measures has proved to be effective and paramount to control ASF spread, at least within the wild boar compartment and probably also for the domestic pig in a situation with such a high wild boar-domestic pig contact as the one reported in this study.

Page 3: since it has not been done, in page 3 (last paragraph before the “Alternative management scenarios) I miss the percentage of wild boar cells where infection came from “additional forces of infection not otherwise captured”.

Alternative management scenarios: The first paragraph of this subsection (end of page 3 and first line of page 4, “Alternative control strategies...were all examined”) is a description of the methodology and, as such, does not belong in Results but in the corresponding section. The remaining four paragraphs of this subsection contain relevant information, but are somehow redundant with figure 4 and table 2. The authors should rewrite this section and redesign figure 4 and table 2 to avoid the repetition and overlap of the information provided, eventually transferring figure 4 to supplementary material if deemed necessary but redundant.

Figure 3 is visual but little informative, due to the long whiskers not allowing to reach a clear understanding of the differences due to each alternative control strategies. Instead, a Table with the numerical values would probably be more informative. Values should be arranged first according to the host (i.e., the five scenarios for domestic pigs on one side and the five scenarios for wild boar on another side) to allow an easier comparison of the effect of the alternative management measures in each host. Similarly for figure 4, which is probably redundant with table 2 (means and IQR from figure 4 could probably be simply added to table 2).

DISCUSSION

The Discussion section is excessively lengthy and detailed. The whole section could and should be rearranged, reordered, and rewritten. The first part justifies the values used and found in relation to previous publications, discussing them one by one. Then the “Alternative management scenarios” subsection within the discussion sections dedicates five paragraphs (one for each alternative management measure analysed) to separately discuss each alternative management scenario. These two sections provide a detailed discussion for each one of the items mentioned, comparing them with previously published values and discussing the potential reasons for every one and each of the effects and results found. While I appreciate the honesty of the authors trying to present and justify their results, demonstrating that the model adequately captures reality, all these arguments add or confirm previously published knowledge, generating little advance. Instead, trying to summarize the main findings and clearly stating the advance in knowledge provided by their model making their manuscript an article worth publication in Nature communications would probably give more value to the Discussion section, at should therefore be placed at the very beginning of this section. The authors try to give some hint of these ideas in the subsection “Policy impact and real-time support”, but I still would like to see clearly stated the advantage of the model with regards to other models and/or epidemics descriptions from other countries previously published.

There is some repetition of results and materials and methods in the discussion section that unnecessarily enlengthens the manuscript (see, for example, at the end of page 5, “Approximately 21% of domestic pig infections were estimated to come from wild boar sources, and roughly 32% of wild boar infections were estimated to originate from domestic pig sources”, which is a literal repetition of results; or, in page 6, “Following determination of the relative host contributions to the overall epidemic, alternate management scenarios were explored”, which is a repetition of materials and methods). These repetitions should be pruned throughout the discussion section and the whole section rewritten and restructured for shortness, clarity, and hierarchical presentation of the main findings.

Page 6 first and second paragraphs of the “Alternative management scenarios” subsection: “Strategies that employed village-wide culling—either preventively upon nearby wild boar case detection ...” and “In addition to reactive culling of pig herds, preventive culling following nearby wild boar case detection was similarly effective at decreasing final epidemic size”. Here concepts are repeated, as a demonstrative example of the poor arrangement of the ideas in the Discussion section, which could be improved in order to shorten the manuscript and make reading and understanding of the more relevant

findings more straightforward.

For example, the third paragraph in page 8 (“The dynamics seen with...distribution and spatial correlation”) provides a nice ASF context review but is only remotely related to the study.

Page 9, model data and limitations: As aforementioned, using more precise and accurate wild boar population estimate data available would surely improve the model, as would also do covering a one-year simulation period.

Page 9: “Indeed, of the 53 livestock diseases in the United States of America that are reportable to WOA, 42 have a wildlife component, with similar livestock-wildlife disease risk and susceptibility seen in Europe (65, 66)”. I do not understand this citation to the United States of America as a reference for Europe, when the same information is available for a European context.

FORMAL COMMENTS

Page 2: “Of the 15 parameters in the model, four were able to be informed from available data in the literature or observed data (Table 1)”. In Table 1 there are 14 parameters. Moreover, the immediately following paragraph starts with “The remaining 110 parameters ...”, and four plus ten add fourteen.

Some conceptual epidemiological misunderstanding can be inferred from the terms used in the manuscript. The authors repeatedly use “spill-over” for ASF transmission between wild boar and domestic pigs in both directions. However, spill-over means that, after the virus has been transmitted from one host type to the other, the infection does not spread in the newly infected host population, and this is clearly not the case for ASF. Instead, “interspecific transmission” should be used. I recommend the authors to carefully revise all the epidemiological terminology used throughout the manuscript to avoid such misunderstanding of concepts.

The abbreviation MSE for mean squared error is defined in Materials and methods. However, since in the format of the manuscript the Results precede Materials and methods, this abbreviation should be defined the first time it appears in the text in sequential order, as the figures for the Materials and methods section are placed after the figures for results.

Throughout the manuscript there is an abuse of subjective, semi-quantitative, non-scientific and/or meaningless wording, such as “many”, “important”, “some”, “In many countries”, several “between” that should be “among”, avoid “very few”, “Indeed”, “Noteworthy”, ... Please address that and try to communicate in a scientific, objective, meaningful writing.

Some examples:

Introduction: “in regards of societal and economical aspects”, instead “according to societal and economical determinants”.

Introduction, page 3, first line: “quantifying the parameters” and not “quantifying parameters”

Page 3: “The external force of infection, representing the mean of all additional forces of infection not otherwise captured, accounted for very few infections to domestic pig units (8.69%, 4.62–15.3%)” should be rewritten as “The external force of infection, representing the mean of all additional forces of infection not otherwise captured, accounted for 8.69% (4.62–15.3%) of the infections to domestic pig units. What significant meaning does “very few” add?

Page 3: “Furthermore, it was found that habitats with a forest coverage greater than 15% were 2.6 times more infectious and 5.3 times more susceptible than habitats with insufficient forest coverage” should be rewritten as “The habitats with a forest coverage greater than 15% were 2.6 times more infectious and 5.3 times more susceptible than habitats with insufficient forest coverage”. What significant meaning does “Furthermore, it was found that” add?

Page 4: “30-week” and not “30 week”

Past verbal tense should be used consistently in the Discussion section to report and discuss the results of the study. Instead, the authors occasionally use the present tense to report and discuss their results.

Page 7: “milieu” is used in its French meaning and not with its English meaning. “variety”, “range”, “diversity”, ... should be used instead,

Page 7: “Noteworthy, this approach was recently employed by a team in the ASF Modelling Challenge (47, 48)”. Does this sentence add any significant meaning related to the results of the study?

Define “EU” the first time this abbreviation is mentioned in the text.

Materials and Methods, Data, second line (page 9): “as a proxy” rather than “to approximate”

And so on.

(Remarks on code availability)

Version 1:

Reviewer comments:

Reviewer #4

(Remarks to the Author)

I commend the authors for their thorough response to reviewer comments and can see that the authors have restructured their manuscript, adapted their methods and provided a more detailed description, and condensed their discussion to more succinctly outline the significance of their work.

Specific responses to reviewers

Reviewer 1 — Interpretation Comment 4

The peak is still not well captured in Tulcea. While I do not believe this is a barrier to publication, I do not feel that Line 205 (While our model could replicate spatial and temporal epidemic trends...) is fully accurate without some caveats. I would like to see the discrepancy commented on here, with potential explanations.

Reviewer 1 — Figures and Tables Comment 2b

I agree with the reviewer that it would still be helpful to present the new Figure 1 with a selection of the best individual trajectories overlaid to understand if the model is capable of matching peaks across all counties in a stochastic simulation.

Reviewer 1 — Interpretation Comment 5

The authors go to significant effort to fit 256 models for their model selection process that is described in detail in the work. After selecting the best model, it appears that 255 model variants are the immediately discarded with no results presented on these variants. It is unclear how similarly the alternative model variants performed. I would like to see a supplementary figure ranking the top model variants by the Euclidean distance to the observed summary statistics and potentially a weighted ensemble of these models for some of the key figures.

Reviewer 1 — Methodology Comment 12

The alternative control strategies show close to 50% probability of having a positive effect and thus they all appear to have a very weak effect. Do the authors believe this to be accurate? I see this is commented on in the discussion and provides some explanation. While not a required recommendation, it would be interesting to isolate the impact of the control strategy using an alternative method such as the Sellke construction or methods described in Sunnucks et al (2025) (<https://www.medrxiv.org/content/10.1101/2025.10.09.25337145v1>). This would help to reduce the stochastic uncertainty and focus on the true impact of the intervention.

Reviewer 3 — “More importantly, the authors neither model external sources of infection for wild boar...”

Similarly to Reviewer 1 Interpretation Comment 5, I would like to understand the difference in including an external force of infection for wild boar. Did the second/third best fitting model include this factor?

Additional comments

Abstract/Discussion

The headline results in the abstract show huge uncertainty, where in some cases the credible interval covers almost the entire range. This should be commented on in the discussion.

Figure S2

The caption states that “patch-to-farm transmission mode and the form of the external force of infection onto patches had the least influence.”

Is this true? It appears that the the external force of infection onto patches has the 4th most influence?

Line 114

“hi- medium- and low-“ should be “high- medium- and low-”

(Remarks on code availability)

It is clear that significant improvements have been made to the codebase inline with reviewer comments.

Version 2:

Reviewer comments:

Reviewer #4

(Remarks to the Author)

The authors have addressed my concerns.

(Remarks on code availability)

Dear Reviewers,

Thank you for taking the time to have reviewed our manuscript, and for providing the considerable detailed, actionable guidance necessary to improve both its content and delivery. In short, following your recommendations, the entire manuscript has been restructured and rewritten, and the entire codebase rebuilt (and confirmed functional on a clean system). Please find below our responses to all your individual remarks.

Reviewer #1 (Remarks to the Author):

I am writing upon reviewing the article entitled 'A multi-host mechanistic model of African swine fever emergence and control in Romania' by Hayes et al.. Clearly a lot of technical work has gone into this study, however I found that the article and supplied code obstructed my ability to assess (i) what the noteworthy results were, (ii) if the work is of significance to the field and (iii) whether the work support the conclusions and claims of the study. I further had trouble running the code on the online repository (as detailed in 'Remarks on code availability') and did not find the methodologies detailed enough for a reader to be able to reproduce the study. Substantial alterations to the manuscript would need to be made to improve it. In the future, I would also request that the authors add line numbering to their manuscript to make it easier for reviewers to give their comments. Below I have outlined the improvements that could be made throughout the manuscript.

Thank you for giving your time to thoroughly review our manuscript and for providing these detailed recommendations for improvement. In-line with your comments, substantial alteration has been made to both the manuscript and the codebase, in the following summary:

- The manuscript has been fully re-written, now presenting material in a hierarchical, concise manner.
- The noteworthy results are better-contextualized in the Abstract (L17–21 "Suspected interspecific transmission... remains poorly understood... here we quantify these dynamics") and Discussion (L160–164).
- The significance of the results to the field is now explicitly stated in the Discussion (L152–159).
- The codebase has been completely refactored, a now includes a novel and streamlined implementation pipeline, removal of the depreciated *rgeos* dependency that broke the spatial component of the model, and clear internal documentation.
- The methodology has been improved (i.e. statistical justifications of model structural choices, exploration of all combinations of model structural variants) and the re-written Methods section provides adequate detail to reproduce the study (e.g. all equations are explicitly provided, and exact definitions of processes [i.e. Tau leap] are stated).
- Line numbers are now included.

INTRODUCTION

1. The introduction states that 'interrupting disease transmission is reliant on control strategies targeting these pathways' and later, 'evaluating potential outcomes from alternative control strategies'. Are the alternative control strategies that are evaluated in the model different from the current controls? It would be useful for the reader if more detail is included on what the current controls are, and how the models evaluate an 'alternative'.

Thank you for this opportunity to improve the reader's situational awareness. To maintain clarity between the observed situation and model hypotheticals, existing control measures are now discussed in the introduction (EU-level strategies on L42–51, and what was implemented in Romania on L72–78). The alternative control strategies, which are hypothetical and explored in the model, are then presented in the first paragraph of the Results after the introduction (L94–97).

RESULTS

1. The methodologies are understandably important, and owed to the format of the journal (i.e. Methods at the end), it would be beneficial if the results had a much gentler first paragraph covering what does the model attempt to capture, the differences between the fitted models, and what kind of data was it fitted to (away from technical jargon).

The opening paragraph of the Results section, *model design* (L86–97), has been re-written to provide a soft introduction to the methods that expresses: 1) the research objective (L86. "capture the spread of ASF in a multi-host system"), 2) general model differences (L89–91. "A total of 256 model formulations... representing all combinations of... model frameworks, transmission formulations and contact structures"), and 3) the data used to fit the epidemic (L88. "calibrated to weekly case detection data")

2. In the first paragraph of results, you should define 'MSE' given that the Methods come later.

3. Was a particular method used to determine that the best fitting model's prediction intervals 'satisfactorily' captured the weekly incidence for each host?

MSE is no longer employed, and model selection criteria are now defined in the first paragraph to explain how epidemic capture was defined (L91–94. "The best-fitting model was defined as the first among the top-ranked candidate models (based on the lowest total distance to the observed summary statistics) where the observed weekly incidence of the majority of counties fell entirely within the 95% credible intervals of the model's posterior predictions.").

4. The work that has gone into fitting all 24 models and estimating uncertainty is substantial. To that end, in the last 2 sentences at the end of paragraph 3 of the Results, can the uncertainty of the time estimates also be supplied in addition to the median?

The results have been restructured so that parameter estimates (along with their conversion from units-per-week to weeks-per-unit as previously provided) are no longer discussed in the main body text, and the reader is instead referred to Table 1 for parameter medians and uncertainty. Indeed, as it had existed previously, this paragraph was confusing and did not benefit the message.

5. Related to the above point, can you please maintain consistency between the intervals presented: throughout there is a mixture between only a point estimate, the inter-quartile range, a 95% equal-tailed interval, and a 95% highest-density interval.

All presentations of intervals (L128–134, Figure 2, Table 1 and Table 2), have been harmonized to 95% credible intervals. IQRs and HDIs are no longer used nor relevant to the presented work.

6. Within the results section, can you explicitly add into each subsection how many parameter samples and stochastic simulations were used to generate of the summary statistics?

Parameter sample and simulation counts are now provided in both the main text (L122–123) and header for Table 1 that provides the parameter estimates.

INTERPRETATION

The discussion is far too long which makes it very challenging to consolidate the study. Mainly, the length obstructs the reader in determining what the noteworthy results are, and if they are of significance to the field.

Thank you for the detailed recommendations on improving the discussion and interpretation. The major changes include a complete restructuring, rewriting, and condensation of the Discussion section, with noteworthy results and advancements to previous studies now explicitly stated in the first two paragraphs (L148–159),

1. Please collapse the discussion down significantly.

We understand how the original discussion was overly long. In the revised manuscript, the discussion has been reorganized and rewritten, with its length reduced by 66% (from 3446 words to 1165 words) while maintaining a cohesive message and removing excessive contextualization.

2. The paragraphs in the discussion are not consistent: start with what your findings were for each section, and then compare these to other studies.

The structure of all paragraphs in the discussion has been harmonized to a consistent format. Now, each paragraph begins with a clear statement of the relevant finding and is then followed by contextualization, such as:

- L160–162. "Our results demonstrated that while domestic pigs and wild boar each sustained their own epidemics, interspecific transmission played a non-negligible role... These estimates were aligned with evidence from South Korea²⁶...";
- L174. "Wild boar spread was more rapid than the velocities observed in Poland and the Baltic states...^{2,3,29,30}";
- L185–186. "Relying on forest coverage as a habitat suitability proxy rather than density estimates resulted in improved transmission modelling among wild boar, reflecting established ecology.^{32,33}";

3. Please adjust the final sentence of the first paragraph in the Discussion reading 'The final result was a model where the overall trends observed in the epidemic emergence period were able to be replicated through stochastic simulation.' It needs to be made more clear that only the 'the spatial distribution of the epidemic is further replicated in the majority of areas', and not all areas.

Thank you for this recommendation, we agree that the epidemic was not as well-captured as it could have been. In the revised study, the capture of spatiotemporal trends is much-improved and the revised discussion makes it clear that the models attempt to replicate the epidemic occurs under simplified conditions (L205–212).

4. There are spikes in domestic pig cases in Braila and Tulcea not explained by the model (Results paragraph 2 & figure S2). What factors not included in the model could explain the inability to capture the spike in weekly incidence in Braila and Tulcea?

In the refactored model that employs updated transmission mechanisms, the alignment between the observed and simulated incidence are improved in both Braila (with peak incidence captured) and Tulcea (with peak incidence nearly-captured). Through modifying the model mechanisms to allow separate farm-to-farm transmission rates for high- medium- and low-incidence counties (with Braila and Tulcea being high-incidence), the estimated parameters could now better-reproduce the observed patterns. Tulcea is still not fully captured, and this likely reflects a combination of unreported animal movements, enhanced connectivity between villages, and/or reporting delays (as this was the first county infected). As the patterns are more-closely reproduced, this aspect was omitted from the discussion to maintain conciseness (instead focusing on the limitation of excluding within-village dynamics [L205–220]), though can be explored further in the revised version if recommended.

5. It would be interesting to see how estimated model parameters differ depending on the twenty-four model designs.

There are now 256 model variation that vary systematically among design elements. We agree it would be interesting to see how these design choices influence model performance, and so both a permutation feature importance of structural parameters was performed, along with an examination of the ECDF of the final model distance by design element (Results: L105–111, Figure S2 and Figure S3, Methods: L407–412).

METHODOLOGY

The study would not be able to be replicated given the level of detail as it stands in the Methods and does not fulfil the standard as expected in the field. Below are some points that could help improve this section:

Thank you for this perspective. In the revised manuscript, the methodology has been rewritten to ensure both transparency and reproducibility, following the recommendations provided below.

1. Please consider rewording the first few sentences of the model agents section. Referring to the models in domestic pig herds and wild boar cells before they are defined is confusing.

The Methods have been fully rewritten, with the *Data and model agents* subsection now opening with a description of each dataset used, followed by distinct paragraphs dedicated to describing the domestic pig farm (L249–254) and wild boar (L255–259) components for clarity.

2. Please be more explicit in defining the compartmentalisation of the model.

The infection state compartmentalization is now presented in its own dedicated paragraph (L281–288), where we describe the modified SIR structure used and the inclusion of the surveillance process through partitioning the infectious state into detected and undetected components, and which units transit through which infection states. A more detailed enumeration of each compartment (and corresponding flow diagram) was not added as we felt the revised manuscript now provides sufficient clarity for readers familiar with infectious disease models. If more detail is still requested however, it can be added to either the main text or supplemental material.

3. Given the forces of infection, the full set of transitions needs to be defined mathematically.

The full set of transitions is now mathematically defined, with each equation numbered and including variable definitions. Formulae for total force of infection, pathway specific forces, the external force of infection, and the transition probabilities for infection, detection, and recovery are now explicitly stated (L281–336).

4. The tau-leap algorithm needs to be more explicitly defined as multiple methods are accepted/discussed in practice, not just the classical one.

The transition algorithm is now explicitly described, where we specify that we use the classical τ -leap approximation where discrete-time transition probabilities are derived from exponential waiting times and applied at fixed intervals (L313–315).

5. The reference (10) also does not refer the appropriate section of the book.

Thank you for catching this error, the appropriate chapter relevant to τ -leap formulations (Stochastic Dynamics) is now cited.

6. Please avoid the explicit use of '*' to indicate scalar multiplication.

All equations (L293, 299, 309, 316, 326, 330) have been reformatted to accepted mathematical notation.

7. Why was the simulation initialised 8 weeks prior to the detection of the first case?

Thank you for identifying this area of obscurity. This decision has been clarified through the inclusion of a distinct subsection *Model initialization* (L367–371), where we explain the timing decision was to account for the estimated carcass detection delay in wild boar, thereby enabling our model to align the first wild boar case detections with the observed data.

8. All 24 models need to be explicitly described, in particular how it would explicitly change the equations of the model. A table would be very helpful here to allow the reader to quickly compare the models.

The set of structural variants has been expanded to yield 256 models, which are now all described in their own subsection *Model structure and variant formulations* (L338–365). Additionally, we have included a table in the supplementary material that summarizes the equation modifications for each component variation to facilitate rapid comparisons between models (Table S1. Alternative formulations of key structural parameters).

9. It's unclear how many parameters there actually are in the model. The Methods states 15, Table 1 indicates 12 and a further 24 depending on the village and quarter. Do the number of parameters also change with the model design?

The presentation of parameters in the original manuscript was indeed confusing, and the manuscript has since been revised to clarify the parameters used in model array. The Methods subsection *Model structure and variant formulations* (L338–365) explicitly describes all structural variations and how it affects the final number of parameters included in a given model. It is now clearly stated that, depending on the structure, a model may contain between 9 and 13

parameters (L362–364), and that the best-fitting model used 11 parameters (L113). Further, this value is consistent with what is reported in the corresponding parameterization table Table 1.

10. In the ABC, how were the 56 summary statistics used coerced into a single value describing distance between the model and data?

We now use 432 summary statistics, representing the weekly incidence per county per host (two hosts, six counties, 36 weeks). The process of variance normalization (to prevent overweighting) followed by taking the maximum absolute distance across all summary statistics is now explicitly described (L393–397).

11. Why was MSE used to determine model selection, and why was it done on a scale completely different to the summary statistics used to fit the models?

We have removed MSE from the model selection process, rather using the same summary statistic distance metric as in the calibration phase to ensure consistency.

12. It's not clear how control outcomes were compared with the baseline outcomes as there are several options particularly given it's a stochastic model.

The approach used to compare control strategies is now explicitly described in the *Baseline and alternative management scenarios* subsection (L420–432). Each scenario was simulated using 500 runs (5 replicates of each of the 100 posterior particles), and the results were summarized by the median and 95% credible intervals of the final epidemic sizes.

13. This applies to summary metrics as well as the model fitting, given that you have a stochastic model, how many realisations of the model were there per parameter set?

Calibration and simulation efforts are now explicitly reported in both the Results (L122–123. 2600 total simulations to obtain a final posterior sample size of 100 particles) and Methods (L391. 200 parameter draws per calibration step).

FIGURES AND TABLES

1. Please add a legend to Figure 1.

A full legend has been included with our new Figure 1 to define both the points and ribbons.

2. Figure 1 is not convincing that the model can replicate the observed dynamics of domestic pigs. It is unclear if this is because of the choice of visualisation, i.e. the summary metrics per time unit aggregate any model-specific noise leading to a non-noisy median, or if the model itself is unable to produce generally noisy trajectories.

To better demonstrate the performance of our model, Figure 1 has been updated to show both overall and disaggregated-by-county model fit—defined through host-specific weekly incidence—and a new supplementary Figure S5 has been included to show the model performance against observed data for all 432 summary statistics.

Not only does Figure S2 more appropriately summarise model performance, it would be good to also show:

a. model predictions against the data at the level of summary statistics (i.e. quarterly incidence per county and per entire study region for domestic pig units and wild boar cells).

As requested, model predictions against the summary data are now presented for both hosts by county in Figure S5, showing both median values and uncertainty intervals.

b. a typical set of trajectories with a low number of replicates.

The revised Figure 1 now shows the 95% prediction ribbons generated from 500 simulations of the best-fit model, to provide a visualization of the model's central tendency and variability of epidemic trajectories. Given that multiple new supplemental visuals have already been added to describe the new facets of the work performed, an additional plot further investigating epidemic trajectories has not been included. If it is felt that a demonstration of raw simulation trajectories would still be beneficial to the communication of the research performed, it can be added as another figure in the supplemental material.

3. Please add a legend to Figure 2 and 4 to define the points and error bars (i.e. median and IQR).

The new Figure 2 has been updated to now define the points and error bars in the caption (L598–604), and figure 4 has since been removed.

4. Table 2 can you please put that these are percentages in the table.

The old table 2 (median reductions in case counts per control strategy) has been replaced with a new table that does not use percentages.

5. An expanded table (as in Figure S1) is needed to describe all 24 models.

The old Figure S1 showcasing the 24 original models has been removed, and the 256 model variations in the new manuscript version are now described in supplementary material Table S1. The new table showcases the model component, the transmission pathway affected, the alternative pathway structures, and the specific equation modifications for each formulation alternative.

6. Supplementary figures showing the model predictions against the data for all other 23 models would also be useful to determine how valuable the model design is in qualitatively capturing the observed dynamics.

Given that the number of model variants has been expanded from 24 to 256, presenting time-series performances of each model would now be impractical. Rather, to demonstrate the influence of model design, we have included new supplementary material Figures S2 and S3 that provide an interpretable summary of the influence of model structure: The relative contribution of each structural parameter to model fit is presented in Figure S2, and the effect of each structural parameter variant on model fit is presented in Figure S3.

7. Table 1 needs a samples size for the 95% credible interval.

Sample size has been included in the title of Table 1.

TYPOS

page 2 'one of the highest consequence diseases': should this be 'one of the high-consequence diseases' since this refers to a class of diseases.

Thank you for this detail, the phrasing has been appropriately updated (L33. "... (ASF), a high-consequence disease of domestic pigs,...").

page 3 'We estimate that a susceptible village was infected': should this be 'We estimated that a susceptible village was infected'.

This sentence has since been removed, and the use of past tense has been harmonized throughout the manuscript.

Reviewer #1 (Remarks on code availability):

Unfortunately, not all required content was marked as complete on the Code and Software Submission Checklist. As a result, it was challenging to spot-check the supplied R scripts and verify some of the codes outcomes. I would suggest getting a colleague to try this software on a brand new machine to test its portability and if the information supplied in the code repository is sufficient.

Below I have outlined some of the features that would be helpful to include in the README file (in addition to features unchecked in the Code and Software Submission Checklist) and stated the issues encountered while running the code in order to help the authors' address these issues. An explicit demo file would be extremely useful to satisfy the execution criteria of the Code and Software Submission Checklist.

My apologies for the technical deficiencies in the original codebase. Following (extensive) improvement in my own programmatic capabilities, the entire model pipeline has been refactored and core code rebuilt to align with both the Nature Research Code and Software Submission Checklist as well as with the recommendations provided.

ADDITIONS FOR THE README.TXT

1. The R version and all package versions that were used to generate the results should be included in at least the main README.txt file, or ideally both README.txt files in each subdirectory '/asf-rom-build' and '/asf-rom', as from sessionInfo(). For example:

```
R version 4.3.2 (2023-10-31 ucrt)
Platform: x86_64-w64-mingw32/x64 (64-bit)
Running under: Windows 11 x64 (build 22631)
```

attached base packages:

```
[1] parallel stats graphics grDevices utils datasets methods base
```

other attached packages:

```
[1] doParallel_1.0.17 iterators_1.0.14 foreach_1.5.2 stringi_1.8.2 janitor_2.2.0
[6] sf_1.0-15 lubridate_1.9.3 forcats_1.0.0 stringr_1.5.1 dplyr_1.1.4
[11] purrr_1.0.2 readr_2.1.4 tidyr_1.3.0 tibble_3.2.1 ggplot2_3.4.4
[16] tidyverse_2.0.0 plyr_1.8.9
```

loaded via a namespace (and not attached):

[1] dotCall64_1.1-1 gtable_0.3.4 spam_2.10-0 raster_3.6-26 htmlwidgets_1.6.4
[6] lattice_0.22-5 tzdb_0.4.0 vctrs_0.6.5 tools_4.3.2 crosstalk_1.2.1
[11] generics_0.1.3 proxy_0.4-27 fansi_1.0.6 pkgconfig_2.0.3 KernSmooth_2.23-22
[16] RColorBrewer_1.1-3 leaflet_2.2.1 lifecycle_1.0.4 compiler_4.3.2 fields_15.2
[21] rgeos_0.6-4 munsell_0.5.0 terra_1.7-65 codetools_0.2-19 leafsync_0.1.0
[26] snakecase_0.11.1 stars_0.6-4 htmltools_0.5.7 maps_3.4.2 class_7.3-22
[31] pillar_1.9.0 classInt_0.4-10 lwgeom_0.2-13 abind_1.4-5 tidyselect_1.2.0
[36] digest_0.6.33 fastmap_1.1.1 grid_4.3.2 colorspace_2.1-0 cli_3.6.2
[41] magrittr_2.0.3 base64enc_0.1-3 dichromat_2.0-0.1 XML_3.99-0.16 utf8_1.2.4
[46] leafem_0.2.3 e1071_1.7-14 withr_2.5.2 scales_1.3.0 2.1-2
[51] timechange_0.2.0 RANN_2.6.1 hms_1.1.3 png_0.1-8 tmaptools_3.1-1
[56] tmap_3.3-4 viridisLite_0.4.2 rlang_1.1.2 Rcpp_1.0.11 glue_1.6.2
[61] DBI_1.1.3 rstudioapi_0.15.0 R6_2.5.1 units_0.8-5

R version and packages are now included in the README file, in the manner above.

2. Typical execution time for one of the shell scripts in '/asf-rom' to fully execute.

Execution times for each model step are now supplied in the README file.

3. Note that one package (rgeos) has been archived on CRAN as since development, but can still be downloaded and installed as of writing.

The rgeos dependency has since been removed, and a full list of required R packages is now provided in the README.

4. Line 102 of '/asf-rom/README.txt' should read 'Bash script "submit-run-par-test-*.sh"' not "'submit-run-par-est-*.sh"'?

This structure is no longer relevant since improving the model implementation pipeline.

5. Please split up SCRIPT FLOW section of '/asf-rom/README.txt' into (i) the commands the user needs to execute to replicate the study, and (ii) what each script does.

The execution steps for running both the full model on a computing cluster as well as the toy model either remotely or locally are now explicitly provided in the new README file.

EXECUTION ERRORS IN CODE

With the refactored implementation pipeline the below errors no longer occur. This has been confirmed through testing both on the development environment and through a clean installation on a separate system.

1. Test of '/asf-rom-build/01-init_data.R' unsuccessful:

a. After changing all hard-coded working directories in each file, this script initially this crashed my machine.

b. Upon removing parallelism (please don't hard-code #nodes to use in a parallel loop in the future), I encountered this error which I assumes comes from f-bin_clc.R#21:

Error in { :
task 1 failed - "cannot derive coordinates from non-numeric matrix"

2. Test of '/asf-rom/bash/submit-run-par-est-412.sh' unsuccessful:

a. After changing working directories etc., the script failed on line 96 with
Error in SimulateModel(I0, pars = priors, fixed.pars, herdDist, unitData, :
argument "dt" is missing, with no default

Called from: SimulateModel(I0, pars = priors, fixed.pars, herdDist, unitData,
dt, maxDist, trans.dist.function = trans.dist.function, dpdp.trans.mode = dpdp.trans.mode,
wbdp.trans.mode = wbdp.trans.mode)

b. With some quick debugging, I can see that dt is defined in the global environment from the
call to init.model, but cannot be seen within the full traceback of the error:

Error in SimulateModel(I0, pars = priors, fixed.pars, herdDist, unitData, :
argument "dt" is missing, with no default

12. SimulateModel(I0, pars = priors, fixed.pars, herdDist, unitData, dt, maxDist, trans.dist.function
= trans.dist.function, dpdp.trans.mode = dpdp.trans.mode,
wbdp.trans.mode = wbdp.trans.mode) at function-par-est.R#36

11. old_model(param_with_constants)

10. model(param)

9. .ABC_rejection_lhs(model, prior, prior_test, nb_simul, use_seed,
seed_count)

8. .ABC_Lenormand(model, prior, prior_test, nb_simul, summary_stat_target,
use_seed, dist_weights = dist_weights, verbose, ...)

7. .ABC_sequential(method, model, prior, prior_test, nb_simul, summary_stat_target,
use_seed, verbose, dist_weights = dist_weights, ...)

6. ABC_sequential(method = "Lenormand", model = model, prior = priors,
nb_simul = n.sim, summary_stat_target = summary_stat_target,
p_acc_min = p.acc.min, verbose = TRUE) at function-par-est.R#60

5. par.est(sub.id, n.sim, p.acc.min, fixed.pars, I0, n.breaks, sum.stats.obs,
sum.stats.target = sum.stats.target, trans.dist.function = trans.dist.function,
dpdp.trans.mode = dpdp.trans.mode, wbdp.trans.mode = wbdp.trans.mode) at run-par-est.R#95

4. eval(ei, envir)

3. eval(ei, envir)

2. withVisible(eval(ei, envir))

1. source("asf-rom/scripts/run-par-est.R")

3. Test of '/asf-rom/bash/submit-run-par-test-408.sh' unsuccessful:

a. The same error occurred as above.

Reviewer #2 (Remarks to the Author):

Overall I found the model to be a pleasant and approachable read, and the authors did a decent
job describing the model and its results, allowing for the difficulty in describing individual based
models, and the journal's "results first" style.

Thank you for this positive commentary.

I have two major points of concern:

1) The combination of the model not being able to model individual herds within a village, as well as the reports discussed by the author of various ways to avoid or evade culling, makes me wonder if the model is overestimating the effectiveness of culling. Some discussion of the sensitivity of the model to incomplete culls is probably not uncalled for.

Thank you for raising this important point. Indeed, in the alternative scenarios of village-wide culling (either reactive or preventive), the assumption of perfect implementation in a system dominated by smallholder farms is unrealistic. In the revised model, only a limited effect on epidemic size was now observed from the alternative culling strategies. We now discuss how treating each village as a homogenous unit likely shaped observed transmission patterns and how this homogeneity assumption resulted in an underrepresentation of the residual reinfection risk (secondary to the suspected individual behaviors of pre-emptive slaughter or swill feeding) (L195–220). Given the diminished impact of culling in the updated results and the removal of any strong conclusions about its effectiveness, we opted not to expand the discussion with a full sensitivity analysis on this aspect. However, we recognize this remains a relevant source of uncertainty and should be explored in future investigations.

2) Some of the parameters in the supplement appear to be practically if not actually non-identifiable. This should be addressed in the body of the paper itself, as this is of major concern.

We agree that certain parameters that contribute weakly or minimally to shaping the (now 432) summary statistics (of incidence per county per host per week) would be practically non-identifiable. The manuscript now explicitly addresses the identifiability of certain parameters in the Results section (L123–125), noting that the seasonal relative infectivity of farms was seen to be only weakly informed by the data, while others like the relative susceptibility of wild boar patches or the farm-to-farm transmission rates were more-clearly identifiable.

Reviewer #2 (Remarks on code availability):

Code appears well documented. Libraries are a little bit cumbersome as tidyverse is installed in its entirety. Code in 01-init_data.R did not run, as the rgeos library appears to be no longer available.

The codebase has been completely rebuilt, and the *tidyverse* suite is no longer loaded in its entirety. Further, the *rgeos* dependency has been removed, and the model has been tested on a clean system to ensure full functionality.

Reviewer #3 (Remarks to the Author):

In the manuscript entitled “A multi-host mechanistic model of African swine fever emergence and control in Romania”, Hayes and collaborator develop a model to assess the variables determining the extension, duration and number of swine affected by an ASF outbreak in Romania, taking into account the specificities of pig production and the wild boar-domestic pig interface in this country. The authors have gathered a reasonable dataset, undertaken a considerable amount of work, and obtained some nice results, but from my point of view they miss to give the article a wider scope, refine their methodology, and synthesize and effectively communicate their most relevant results in a hierarchical and straightforward way. Additionally, the manuscript has a non-negligible number of formal issues that should be addressed. Although the format requirements for Nature communications are rather relaxed, the whole review process would have been much easier and simpler if the manuscript was line-numbered. English needs style and

grammar revision, since part of the length of the manuscript can be attributed to not-so-straightforward language and writing, apart from other errors (see specific comments). While they achieve a relatively acceptable explanation of one country-case and provide some relevant data regarding the percentage of infections in each compartment (i.e., domestic pigs and wild boar) originating from cross-transmissions (which is relevant not only for modelling, but also for biosecurity and management), the Discussion section (and the Policy impact and real-time support within it) fails to communicate an expected impact of the study beyond the adjustment of the model to the description of this one country-case. Most of the Discussion section is focused on justifying the results and comparing them with previous data of other country-cases (see specific comments below), which does not provide a significant advance in knowledge but adds to previously existing literature. Instead, the authors could extend on the applicability and advantages provided by the model.

Other major flaws raising concern include methodological aspects (see specific comments below), with limitations both in the assessment of domestic pig farm infection and, particularly, in the indirect estimation of wild boar population. Refining these two aspects would probably produce more accurate results, as would do extending their modelling period to a complete natural year.

We thank the Reviewer for the thorough critique of our article along with actionable recommendations to help our manuscript not just add to existing literature but provide an advancement to the field. In response, the model codebase has been fully rebuilt and the manuscript nearly-completely rewritten, having been restructured in a hierarchical manner with revisions designed to improve the generality, clarity, and methodological reproducibility. In summary of the point-by-point responses that follow:

- The Discussion section has been rewritten to emphasize the broader relevance and policy implications of our findings, as well as highlight the novel contributions of our approach.
- The methodology has been expanded to explore the full set of alternative model structural formulations (256 combinations, up from 24), and now includes explicit data-informed justifications of major methodological decisions, including having explored the use of both forest coverage and ENETWILD wild boar density estimates to inform habitat suitability as suggested.
- Line numbers have been included to facilitate review.
- The study period has been better contextualized, with clear justification provided for our focus on the June–December 2018 period: This period was identified as a distinct epidemic wave aligned to ASF's initial invasion and early establishment in Romania.

SPECIFIC COMMENTS

Methodology: some criticism could be made to the design and construction of the variables, i.e., assuming villages as a single epidemiological unit; giving the same relevance to all the villages and industrial farming units without considering relevant factors for ASF transmission such as biosecurity and number of pigs per village/industrial farm; ... for example, the number of pigs per intensive farms could be included in the density-dependence of transmission within the model, although admittedly, since intensive breeding pig farms are relatively scarce and supposedly more biosafe than backyard pigs, this would probably not affect significantly the outcome. However, any other option could also be criticized, so the decisions of the authors can be also valid. The main concern remains about the depth of the information used to construct the model, since more detailed quantitative and qualitative information for the variables used would probably have refined the output, without need to add further variables. The categorical proxy for

estimating wild boar population abundance (i.e., above of below 15% of forest over) could be improved, since finer and more detailed data on wild boar population estimates can be achieved through existent models previously developed for Europe, including Romania (see, for example, ENETWILD consortium et al. 2019, 2021). Such models are also available to estimate the wild boar-domestic livestock (including swine) interface (ENETWILD consortium et al. 2020). The fact that “all models that used density-dependent distance kernels or frequency-dependent step functions performed better than those that used a density-dependent step function” (page 3) further supports the idea of more complete, precise, and detailed data feeding to the model contributing to improve the accuracy and reliability of the output.

- ENETWILD consortium, Acevedo P, Croft S, Smith G, Blanco-Aguiar JA, Fernández-López J, Scandura M, Apollonio M, Ferroglio E, Keuling O, Sange M, Zanet S, Brivio F, Podgorski T, Petrovic K, Soriguer R and Vicente J, 2019. ENETwild modelling of wild boar distribution and abundance: update of occurrence and hunting data-based models. EFSA supporting publication 2019:EN-1674. 29 pp. doi:10.2903/sp.efsa.2019.EN-1674

- ENETWILD consortium, Fernández-López J, Acevedo P, Blanco-Aguiar JA, Vicente J, 2020. Analysis of wild boar-domestic pig interface in Europe: preliminary analysis. EFSA supporting publication 2020:EN-1834. 45 pp. doi:10.2903/sp.efsa.2020.EN-1834

- ENETWILD consortium, Illanas S, Croft S, Smith GC, Fernández-López J, Vicente J, Blanco-Aguiar JA, Pascual-Rico R, Scandura M, Apollonio M, Ferroglio E, Keuling O, Zanet S, Brivio F, Podgorski T, Plis K, Soriguer RC and Acevedo P, 2021. Update of hunting yield-based data models for wild boar and first models based on occurrence for wild ruminants at European scale. EFSA Supporting Publication 2021:EN-6825. 30pp. doi:10.2903/sp.efsa.2021.EN-6825

Thank you for these detailed and constructive critiques. To assess potential within-village heterogeneity, we explored using the available human census data as a proxy for the number of backyard herds (L272–276). However, only a weak association was found between village size and infectious period duration, suggesting limited explanatory value. Coupled with farm-level pig population data not being available, we elected to retain the aggregate approach and exclude within-village heterogeneity, though instead we acknowledge and discuss this in the revised Discussion (L205–220). For the wild boar population, we have expanded the methodological framework to use both CORINE landcover and ENETwild density-estimate approaches for estimating wild boar distribution (L255–259). After integrating and evaluating both sources, the forest coverage proxy showed superior case predictive performance and was selected for use in the final model set (L199–104).

Page 5: “Research into the social dynamics around pig farming during this time period would help to further clarify the accuracy of this fitted value. As the subsequent epidemic wave in 2019 was initiated at a similar period, it is possible that this temporal consistency is associated with this annual social dynamic, though further investigation is required to confirm this hypothesis”. I have been wondering all throughout the manuscript why the authors selected only a half-year period instead of a year-round period, as most simulation studies do, in order to capture all the seasonality and not just the December slaughter one. Since backyard pig rearing takes place mostly seasonally from July to December (something the reader learns only in the results section), developing a whole year-round model makes perfect sense to capture the epidemiological dynamics of ASF in Romania.

Thank you for this suggestion, we now clarify in the introduction that the modelled period (June–December 2018) was intentionally selected to capture the first full epidemic wave (L67–72). This period exhibited clear onset, exponential growth, and recovery phases (among domestic pigs,

wild boar surveillance was known to be incomplete), which made it well-suited for investigation without requiring additional assumptions (i.e. disease reintroduction mechanisms, or changing transmission dynamics following a mass cull). While we agree that extending the model to explore the epidemic after it had reignited in early 2019 would be of benefit and a natural extension to our work, doing so would likely require expansion of the existing model into a structural framework beyond the scope of this study. However, we recognize this would be a promising direction for expansion of this work.

The authors also dismiss in their model the probability of reinfection in villages. I assume that this would complicate both modelling and output interpretation, but a 13% ratio is not negligible.

We agree that the ratio is not negligible, so we have now further investigated this aspect of our model (L213–220, 260–271). We recognize this value is sensitive to the infectious period definition that we apply, so to better inform this, we tested a range of post-detection infectiousness duration assumptions to identify a threshold that avoids artificially splitting single outbreaks into multiple ones while remaining consistent with plausible outbreak dynamics. Using an elbow-method heuristic, we found that a 3-week window aligned with the experience of the field investigations, which had suggested village-level infectiousness lingers by 2–4 weeks. Applying this value, the number of assumed reinfection events decreases to affect 10% of villages. This proportion is more likely to reflect residual reinfection risk than overestimate recurrence secondary to an artefact in our methods. Additionally, we discuss how certain behaviors can increase re-infection risk, and would be an area for future expansion of the model (L216–220).

More importantly, the authors neither model external sources of infection for wild boar, thus yielding three infection rate parameters for domestic pigs but only two for wild boar. Nevertheless, there is abundant literature demonstrating the RELEVANCE of external sources of wild boar infection for the onset of the epidemics. The cases in Romania (as in Poland, Czech Republic, Germany, Belgium, and Italy) started with the detection of wild boars dead from ASF, therefore originating from such “additional sources of infection not otherwise captured” (page 3). The relevance of such infection sources cannot therefore be neglected.

We appreciate this observation and recognize the importance of considering unexplained transmission events among wild boar beyond the initial seeded infection, so it is now directly addressed in our inference pipeline (L359–361). In the updated model calibration, we explicitly evaluate versions of the model that include an external force of infection acting upon wild boar. In the updated analysis, the best-fitting model did not include this parameter (L113–114), and the implications of this are now discussed (L176–184). With our model accounting for interspecific transmission, it is suspected that many of the long-range drivers of transmission among wild boar are now captured through spillover events following human-mediated long-distance transmission between domestic pigs.

Maybe I have missed it, but the authors mention that the factor regulating the infectivity of domestic pig herds was seasonality, and I have not found in the Materials and methods section a description with the adequate reference for the seasons with higher and lower ASF infection susceptibility. In the results (pages 3 and 5, respectively), the reader learns that it is in winter (“winter holiday slaughter period”) and specifically in December (“December festivals”), but the precise dates, the parameters, and the reasons and references supporting the higher risk are still missing and should be further and better explained in the Materials and methods section.

Thank you for pointing out this area for improving the communication of an important model facet, indeed justification and implementation were not clearly described in the previous manuscript. In the re-written introduction, we now include the relevant social production context to justify accounting for this period (L70–72), and in the Methods we explicitly state the holiday slaughter effect is modulated by the parameters for relative infectivity and susceptibility of farms and is implemented in model week 24, corresponding to early December (L294–298). To avoid confusion, the term “season” has been removed. To clarify, this period is not modelled as one of elevated risk, but rather as one where reduced transmission would be expected to occur secondary to the abrupt decline in the susceptible domestic pig population during the mass cull. The broader implications of accounting for this dynamic are further discussed in the revised Discussion section (L185–194).

“Transmission between herds was further informed by a step function bound at 20 km for the distance between infectious herd i and susceptible herd j (필!,#),” Why 20 km? ASF transmission distance among herds, in absence of pig movements between farms, is supposed to be lower, since ASF is not air-transmitted as CSF but transmitted by close contact, isn’t it? Actually, active ASF surveillance after an outbreak in a farm extends over a 10 km radius, as acknowledged by the authors in the Materials and methods section. How is then supposed an infected herd be infective for another herd at 20 km? At the end of page 7 the reader can find an explanation for the 20-km range (“to capture non-reported pig movements between villages which principally occur at a small scale, our maximum distance was increased accordingly”), but for this assumption to be valid the authors should include it in the Materials and methods section with a reliable reference. Otherwise, as aforementioned, the selection of 20 km, even if intended to account for inter-village commercial pig movements, is random and a non-scientific assumption.

To avoid relying on an arbitrary transmission assumption, in our revised model and manuscript the local transmission distance is set according to the 10 km radius defined in the Romanian legislation. This update is explicitly stated in the Methods (L327–329, 344–345). Further, by keeping the transmission distance aligned to the legislation, we were able to assess whether reinforcing existing policy measures alone would be sufficient to have curbed the epidemic. Our updated findings suggest that without having broadened the scope of control interventions (such as extending the size of the control zones), such measures were unlikely to achieve effective control (L195–204).

In fact, there is a whole paragraph in the Results section (page 3: “Though the distance kernel (핵) for transmission between domestic pig units was estimated as well, the best fitting model utilized a step function (with the force of infection constant at distances shorter than 20 km and null otherwise) to scale the force of infection between domestic units with increasing distance. We estimate that a susceptible village was infected by an infectious village a median of every three weeks, while a susceptible wild boar habitat cell was infected by a neighbouring infectious cell every four weeks. The transmission rate from an infected cell to village herds was estimated at approximately two cases every three weeks, and the transmission rate from village herds to the local wild boar environment was estimated at approximately one case per week”) that is rather Materials and methods. Here the authors are not communicating their results but describing the parameters they chose for the model. This part should not only be transferred to the Materials and methods section, but also adequately referenced (otherwise all the transmission rates selected appear random). Probably after searching and adding the adequate references the authors will consider re-doing some analyses. For example, as aforementioned, a transmission

from village-to-village constant below 20 km and null over 20 km does not look adequately and finely scaled to me. Instead of choosing invariable values for these (and other) parameters to be included in the model, I suggest the authors to carry out sensitivity tests to verify the best-fitting values for the parameters they use to construct the model. This is probably another way to potentially improve the accuracy and predictive value of their model.

Thank you for identifying the confusion from this paragraph. The original intent was to contextualize the inferred parameter by translating the estimated weekly rates (presented in table 1) into more interpretable average times between infection events (e.g. an estimated farm-to-farm weekly transmission rate of 0.307 implies a transmission event occurring approximately every 3 weeks). We now see how this phrasing could have been misinterpreted as describing fixed values used to construct the model, rather than outputs derived during model fitting. To avoid this misunderstanding, we have removed the paragraph from the updated manuscript, left descriptive parameter information to the parameter estimate table (Table 1), and have confined all model construction details to the Methods section.

In the updated model, we assessed sensitivity in two ways. First, we examined the effect of the surveillance zone effect multiplier (a parameter which represented the increase in detection probability within zones surrounding detected cases). and found it had a negligible influence on final epidemic size (L144–146, Table S2). Additionally, we performed a structural sensitivity analysis to determine how alternative modelling assumptions influenced model fit (Methods: L407–412, Results: L105–111),

The authors justify that their best-fit models (figures 1 and S2) capture the actual epidemic trends are included within the 99% interval both for domestic pig units and wild boar cells, either considered altogether or separately by county. However, they fail to capture the minor detail of the epidemic waves, particularly after they reach the peak and the epidemic fades away or turns into endemicity and particularly for the specific county cases.

The 99% credible interval of the scenarios provided by the model are “generous” in forecasting a high spread of the disease, thus it is not surprise that the real epidemic waves fall included within the forecasted prevalence. However, looking at the detailed trend, the correspondence is not so accurate. This inaccuracy improves when the counties are pooled for the whole of the study area (figure 1), but I guess it could be still more refined if increasing the detail of the input variables (quantitative numbers for pig breeding sites, progressive transmission distance between pig farms up to 20 km, finer wild boar population estimations, ... The low effect of environmental sanitation (i.e., removal of ASF-dead wild boar carcasses) proposed by the model in the alternative management scenarios is another evidence of the space for improvement and refinement of the model, since in the real (not simulated) world this measures has proved to be effective and paramount to control ASF spread, at least within the wild boar compartment and probably also for the domestic pig in a situation with such a high wild boar-domestic pig contact as the one reported in this study.

Indeed in our original model there was room for improvement in the capture of county-level epidemic trajectories. In the revised manuscript, the new best-fitting model provides an improved fit that better-captures peaks in high-incidence counties (Figure 1). Additionally, reporting has been revised to 95% credible intervals, which now are used in-standard throughout the manuscript rather than the previous presentation of a variety of metrics. We acknowledge that even in this updated version fine-scale fluctuations in trajectory are not replicated, and case counts among domestic pig farms in low-incidence counties are overestimated. However, this is

likely due to our model being designed for a regional level rather than overfit to local noise, along with not including unobserved contact structures that likely informed such transmission events.

Regarding environmental sanitation, we agree that the measure is a critical component of ASF control. Our model was calibrated to the initial epidemic wave however, during which wild boar surveillance was imperfect and surveillance strength varied between counties. Consequently, the assumption was that many carcasses were likely to have been missed during this early period, and given the short 6-month timeframe of the model, we assumed any removal rate would be too low to achieve environmental sanitation in that period. This would be worthwhile to explore in future model developments especially as finer-scale dynamics are able to be included.

Page 3: since it has not been done, in page 3 (last paragraph before the “Alternative management scenarios) I miss the percentage of wild boar cells where infection came from “additional forces of infection not otherwise captured”.

An external force of infection upon wild boar was not included in the original model, however the revised version now includes one.

Alternative management scenarios: The first paragraph of this subsection (end of page 3 and first line of page 4, “Alternative control strategies...were all examined”) is a description of the methodology and, as such, does not belong in Results but in the corresponding section. The remaining four paragraphs of this subsection contain relevant information, but are somehow redundant with figure 4 and table 2. The authors should rewrite this section and redesign figure 4 and table 2 to avoid the repetition and overlap of the information provided, eventually transferring figure 4 to supplementary material if deemed necessary but redundant.

Figure 3 is visual but little informative, due to the long whiskers not allowing to reach a clear understanding of the differences due to each alternative control strategies. Instead, a Table with the numerical values would probably be more informative. Values should be arranged first according to the host (i.e., the five scenarios for domestic pigs on one side and the five scenarios for wild boar on another side) to allow an easier comparison of the effect of the alternative management measures in each host. Similarly for figure 4, which is probably redundant with table 2 (means and IQR from figure 4 could probably be simply added to table 2).

Thank you for this suggestion for improving the manuscript’s layout and reducing redundancy. In the revised version, methods are kept to the Methods section and results are confined to the Results section (save for a brief methods summary at the start of Results, to provide necessary context while remained aligned with the journal format). Old figures 3 and 4 and table 2, which had presented overlapping information, have been replaced with a streamlined results subsection on alternative management scenarios (L136–142). The new results description references a single updated table (Table 2) that summarizes the outcomes of the scenarios that were explored.

DISCUSSION

The Discussion section is excessively lengthy and detailed. The whole section could and should be rearranged, reordered, and rewritten. The first part justifies the values used and found in relation to previous publications, discussing them one by one. Then the “Alternative management scenarios” subsection within the discussion sections dedicates five paragraphs (one for each

alternative management measure analysed) to separately discuss each alternative management scenario. These two sections provide a detailed discussion for each one of the items mentioned, comparing them with previously published values and discussing the potential reasons for every one and each of the effects and results found. While I appreciate the honesty of the authors trying to present and justify their results, demonstrating that the model adequately captures reality, all these arguments add or confirm previously published knowledge, generating little advance. Instead, trying to summarize the main findings and clearly stating the advance in knowledge provided by their model making their manuscript an article worth publication in Nature communications would probably give more value to the Discussion section, at should therefore be placed at the very beginning of this section. The authors try to give some hint of these ideas in the subsection "Policy impact and real-time support", but I still would like to see clearly stated the advantage of the model with regards to other models and/or epidemics descriptions from other countries previously published.

Thank you for this feedback, the previous version of the Discussion was indeed far too long and disorganized. In response, the Discussion section has been fully rewritten to improve clarity, brevity, and impact, focusing on novel findings rather than contextualizing minor details. The advancement of this work is now clearly stated in the opening paragraph (L149–152), and the second paragraph explains what we feel is the advancement in knowledge that our study contributes to the field (L152–159).

There is some repetition of results and materials and methods in the discussion section that unnecessarily enlengthens the manuscript (see, for example, at the end of page 5, "Approximately 21% of domestic pig infections were estimated to come from wild boar sources, and roughly 32% of wild boar infections were estimated to originate from domestic pig sources", which is a literal repetition of results; or, in page 6, "Following determination of the relative host contributions to the overall epidemic, alternate management scenarios were explored", which is a repetition of materials and methods). These repetitions should be pruned throughout the discussion section and the whole section rewritten and restructured for shortness, clarity, and hierarchical presentation of the main findings.

Page 6 first and second paragraphs of the "Alternative management scenarios" subsection: "Strategies that employed village-wide culling—either preventively upon nearby wild boar case detection ..." and "In addition to reactive culling of pig herds, preventive culling following nearby wild boar case detection was similarly effective at decreasing final epidemic size". Here concepts are repeated, as a demonstrative example of the poor arrangement of the ideas in the Discussion section, which could be improved in order to shorten the manuscript and make reading and understanding of the more relevant findings more straightforward.

Thank you for identifying these areas of redundancy, they have been removed and the updated Discussion repeats neither methods nor results.

For example, the third paragraph in page 8 ("The dynamics seen with...distribution and spatial correlation") provides a nice ASF context review but is only remotely related to the study.

The unnecessary context has been removed in the updated manuscript.

Page 9, model data and limitations: As aforementioned, using more precise and accurate wild

boar population estimate data available would surely improve the model, as would also do covering a one-year simulation period.

The justification for the half-year timeline along with the exploration of multiple wild boar population proxies are now explored, as mentioned in previous comments.

Page 9: "Indeed, of the 53 livestock diseases in the United States of America that are reportable to WOAHA, 42 have a wildlife component, with similar livestock-wildlife disease risk and susceptibility seen in Europe (65, 66)". I do not understand this citation to the United States of America as a reference for Europe, when the same information is available for a European context.

We see how this information about the US is irrelevant to our study, it has been removed.

FORMAL COMMENTS

Page 2: "Of the 15 parameters in the model, four were able to be informed from available data in the literature or observed data (Table 1)". In Table 1 there are 14 parameters. Moreover, the immediately following paragraph starts with "The remaining 110 parameters ...", and four plus ten add fourteen.

The presentation of parameters in the original manuscript was confusing, and a similar comment was raised by Reviewer #1. We have revised the manuscript and clarified the parameterization of the updated model array. Now, the Methods subsection *Model structure and variant formulations* (L338–365) explicitly describes all structural variations and how it affects the final number of parameters included in a given model. It is now clearly stated that, depending on the structure, a model may contain between 9 and 13 parameters (L362–364), and that the best-fitting model used 11 parameters (L113). Further, this value is consistent with what is reported in the corresponding parameterization table, Table 1.

Some conceptual epidemiological misunderstanding can be inferred from the terms used in the manuscript. The authors repeatedly use "spill-over" for ASF transmission between wild boar and domestic pigs in both directions. However, spill-over means that, after the virus has been transmitted from one host type to the other, the infection does not spread in the newly infected host population, and this is clearly not the case for ASF. Instead, "interspecific transmission" should be used. I recommend the authors to carefully revise all the epidemiological terminology used throughout the manuscript to avoid such misunderstanding of concepts.

Thank you for the enlightening insight, all terminology has been revised to ensure that what was previously referred to as spillover is now correctly termed interspecific transmission.

The abbreviation MSE for mean squared error is defined in Materials and methods. However, since in the format of the manuscript the Results precede Materials and methods, this abbreviation should be defined the first time it appears in the text in sequential order, as the figures for the Materials and methods section are placed after the figures for results.

This method is no longer used and the abbreviation has been removed, and all abbreviations are now defined when they first appear.

Throughout the manuscript there is an abuse of subjective, semi-quantitative, non-scientific and/or meaningless wording, such as “many”, “important”, “some”, “In many countries”, several “between” that should be “among”, avoid “very few”, “Indeed”, “Noteworthy”, ... Please address that and try to communicate in a scientific, objective, meaningful writing.

The manuscript language has been updated to reflect the appropriate styling expected of a scientific article, and all the examples given below have been corrected.

Some examples:

Introduction: “in regards of societal and economical aspects”, instead “according to societal and economical determinants”.

This phrasing is now no longer used in the manuscript.

Introduction, page 3, first line: “quantifying the parameters” and not “quantifying parameters”

Definite articles are included where appropriate.

Page 3: “The external force of infection, representing the mean of all additional forces of infection not otherwise captured, accounted for very few infections to domestic pig units (8.69%, 4.62–15.3%)” should be rewritten as “The external force of infection, representing the mean of all additional forces of infection not otherwise captured, accounted for 8.69% (4.62–15.3%) of the infections to domestic pig units. What significant meaning does “very few” add?

Page 3: “Furthermore, it was found that habitats with a forest coverage greater than 15% were 2.6 times more infectious and 5.3 times more susceptible than habitats with insufficient forest coverage” should be rewritten as “The habitats with a forest coverage greater than 15% were 2.6 times more infectious and 5.3 times more susceptible than habitats with insufficient forest coverage”. What significant meaning does “Furthermore, it was found that” add?

Unnecessary verbiage has been removed in the updated manuscript, and these specific paragraph sections are no longer relevant.

Page 4: “30-week” and not “30 week”

Grammar has been corrected throughout.

Past verbal tense should be used consistently in the Discussion section to report and discuss the results of the study. Instead, the authors occasionally use the present tense to report and discuss their results.

The verb tense throughout the manuscript has been corrected, with results always referred to in the past tense.

Page 7: “milieu” is used in its French meaning and not with its English meaning. “variety”, “range”, “diversity”, ... should be used instead,

Language has been standardized to English meanings.

Page 7: "Noteworthy, this approach was recently employed by a team in the ASF Modelling Challenge (47, 48)". Does this sentence add any significant meaning related to the results of the study?

Agreed this does not add anything to the relevance of the results; the sentence has been removed.

Define "EU" the first time this abbreviation is mentioned in the text.

Abbreviations are now defined the first time they appear.

Materials and Methods, Data, second line (page 9): "as a proxy" rather than "to approximate"

Noted, this phrasing has been removed.

REVIEWER COMMENTS

Dear Reviewer #4,

Thank you for taking the time to review our revised manuscript. We recognize the extra labour involved in evaluating a revision when the initial commentary was provided by others, and thank you for the time and effort required for a thorough evaluation. Below please find our responses to all points, along with the location of changes within the manuscript body.

I commend the authors for their thorough response to reviewer comments and can see that the authors have restructured their manuscript, adapted their methods and provided a more detailed description, and condensed their discussion to more succinctly outline the significance of their work.

Specific responses to reviewers

Reviewer 1 — Interpretation Comment 4

The peak is still not well captured in Tulcea. While I do not believe this is a barrier to publication, I do not feel that Line 205 (While our model could replicate spatial and temporal epidemic trends...) is fully accurate without some caveats. I would like to see the discrepancy commented on here, with potential explanations.

We agree the peak is not well-captured in the county of initial detection, and have revised the manuscript accordingly. Instead, we now open the paragraph on L227 with the modification "While our model could **mostly** replicate...", as well as include an additional paragraph (L235–242) discussing the observed discrepancy and potential reasons behind it. Specifically, we surmise that under-ascertainment during epidemic onset followed by rapid local intensification of response measures was at least partially-responsible for the observed peak. As we assumed a fixed detection probability representing the average response over the modelled period, this surge would not have been able to be reproduced without accounting for time-varying surveillance in our model structure.

Reviewer 1 — Figures and Tables Comment 2b

I agree with the reviewer that it would still be helpful to present the new Figure 1 with a selection of the best individual trajectories overlaid to understand if the model is capable of matching peaks across all counties in a stochastic simulation.

The ten best-fitting stochastic trajectories, selected through mean RMSE, are now overlayed on the 95% prediction intervals in Figure 1. To maintain clarity, the overall plots have been faceted by host as well, and the caption been updated to include: "Individual lines indicate trajectories from the ten top-ranked stochastic replicates (selected using residual mean squared error)." (L647–648).

Reviewer 1 — Interpretation Comment 5

The authors go to significant effort to fit 256 models for their model selection process that is described in detail in the work. After selecting the best model, it appears that 255 model variants are the immediately discarded with no results presented on these variants. It is unclear how similarly the alternative model variants performed. I would like to see a supplementary figure ranking the top model variants by the Euclidean distance to the observed summary statistics and potentially a weighted ensemble of these models for some of the key figures.

Thank you for this consideration, we agree that presenting the outcomes of alternative model formulations provides better context for model selection and aids in understanding of structural uncertainty. Models that satisfied fit criteria (as previously defined in: L446–451) were ranked by their Euclidean distance, from which three clusters emerged numerically (Figure S6). Examination

of trajectories of the models in each cluster (Figure S7) revealed that the first two clusters of 12 total models produced similar predictions, and that models in the third more-distant cluster increasingly diverged in timing or magnitude (particularly among the wild boar component). The models in the first two groups were therefore treated as a single cluster, and were used to create the ensemble from which resulting predicted trajectories (Figure S8) and transmission frequency estimates (Figure S9) were derived. This was achieved through performance-related averaging of the outputs from the simulations of each model. Details are presented through additional results (L142–155), methods (L472–483), and supplementary figures (Figures S7–S10).

Reviewer 1 — Methodology Comment 12

The alternative control strategies show close to 50% probability of having a positive effect and thus they all appear to have a very weak effect. Do the authors believe this to be accurate? I see this is commented on in the discussion and provides some explanation. While not a required recommendation, it would be interesting to isolate the impact of the control strategy using an alternative method such as the Sellke construction or methods described in Sunnucks et al (2025) (<https://www.medrxiv.org/content/10.1101/2025.10.09.25337145v1>). This would help to reduce the stochastic uncertainty and focus on the true impact of the intervention.

We appreciate the guidance for augmenting our investigation, and agree that exploring alternative control strategies while employing techniques that remove the Monte Carlo noise in our model can potentially uncover latent effects and provide deeper insights into the effectiveness of the employed strategies. Given the agent-based spatial framework of our model, I do not think that a Sellke construction is able to be feasibly implemented. The method proposed by Sunnucks et al could potentially be implemented (relying on pre-set random number streams to fix stochastic variability), however doing so would require restructuring of the simulation engine enough that it would be more-appropriate as an independent study.

However, we agree that more robust comparison accounting for structural variation is beneficial, so as an alternative, we have extended our ensemble framework to also include the intervention analysis. Performance-based weights were applied to scenario outputs from each of the top models, and are now presented in the main text (results: L156–164; methods: L472–483). The computed ensemble medians and credible intervals for effects on final epidemic size closely mirrored those of the single best-fitting model, suggesting that the estimated effects are stable across model structures and thus less-likely to be driven by stochastic uncertainty.

Reviewer 3 — “More importantly, the authors neither model external sources of infection for wild boar...” Similarly to Reviewer 1 Interpretation Comment 5, I would like to understand the difference in including an external force of infection for wild boar. Did the second/third best fitting model include this factor?

An external force of infection for wild boar was explicitly included in half ($n = 128$) of all tested models. Of the 40 models that fulfilled inclusion criteria, only 8 included an external force of infection for wild boar and they were all in the bottom decile when ranked by Euclidean distance. These models grossly over-predicted epidemic trajectories for both domestic pig and wild boar compartments, and none of them appeared in the two lowest (well-fitting) clusters identified in Figure S7. This suggests that additional unobserved infection processes among wild boar, beyond what is accounted for in patch-to-patch or farm-to-patch, are not supported given our framework.

Additional comments

Abstract/Discussion

The headline results in the abstract show huge uncertainty, where in some cases the credible interval covers almost the entire range. This should be commented on in the discussion.

Thank you for this observation, addressing the wide uncertainty intervals in the transmission frequency estimates will be beneficial for the reader. To better contextualize their interpretation, a new paragraph has been added to the discussion discussing how these estimates reflect both biological and structural uncertainty (L243–251).

Figure S2

The caption states that “patch-to-farm transmission mode and the form of the external force of infection onto patches had the least influence.”

Is this true? It appears that the the external force of infection onto patches has the 4th most influence?

Thank you for catching this error. Indeed the wrong structural parameter was referenced in the caption, and it has been updated to accurately reflect the plot: “Figure S2. Permutation feature importance of structural parameters. Farm-to-farm transmission mode and the contact structure between patches and farms had the greatest impact on final model fit. **Conversely, the interspecific transmission mode between farms and patches, along with the form of the farm-to-farm transmission rate, had the least influence.**”

Line 114

“hi- medium- and low-“ should be “high- medium- and low-”

Grammatical revision has been made, thank you.

Reviewer #4 (Remarks on code availability):

It is clear that significant improvements have been made to the codebase inline with reviewer comments.

Thank you for this acknowledgement, the revisions were substantial but the technical advancements, code fortification, and personal scientific development were well-worth it.